# DISTRIBUTION-FREE FAIR FEDERATED LEARNING WITH SMALL SAMPLES

## ABSTRACT

As federated learning gains increasing importance in real-world applications due to its capacity for decentralized data training, addressing fairness concerns across demographic groups becomes critically important. However, most existing machine learning algorithms for ensuring fairness are designed for centralized data environments and generally require large-sample and distributional assumptions, underscoring the urgent need for fairness techniques adapted for decentralized systems with finite-sample and distribution-free guarantees. To address this issue, this paper introduces FedFaiREE, a post-processing algorithm developed specifically for distribution-free fair learning in decentralized setting with small samples. Our approach accounts for unique challenges in decentralized environments, such as client heterogeneity, communication costs, and small sample sizes frequently encountered in practical applications. We provide rigorous theoretical guarantees for both fairness and accuracy, and our experimental results further provide robust empirical validation of these theoretical claims.

## 1 INTRODUCTION

Federated learning is a machine learning technique that harnesses data from multiple clients to enhance performance. Notably, it accomplishes this without the need to centralize all the data on a single server (McMahan et al., 2017). With the growing integration of Federated Learning in practical applications, *fairness* is gaining prominence, especially in domains like healthcare (Joshi et al., 2022; Antunes et al., 2022) and smartphone technology (Li et al., 2020; Yang et al., 2021). However, applying existing fairness methods directly can be challenging, primarily because many of these methods were originally designed within a centralized framework. This can lead to poor performances or high communication costs when implementing them in real-world scenarios.

To tackle the fairness challenges in the context of federated learning, recent research has introduced several techniques, including FairFed (Ezzeldin et al., 2023), FedFB (Zeng et al., 2021), FCFL (Cui et al., 2021), and AgnosticFair (Du et al., 2021). These methods aim to enhance fairness by implementing debiasing at the local client level and fine-tuning aggregation weights on the server. However, despite their promise, these approaches face certain challenges. Firstly, as highlighted by Hamman & Dutta (2023), achieving global fairness by solely ensuring local fairness can prove elusive. In other words, ensuring fairness for all clients individually may not necessarily result in overall fairness across the federated system. Secondly, many existing methods assume an ideal scenario of infinite samples or struggle to guarantee fairness constraints in a distribution-free manner. This limitation is in contrast to real-world applications. For example, when developing decision models across multiple hospitals or medical institutions, stringent privacy regulations and data access limitations often mean that only limited data can be utilized.

To address these concerns, drawing inspiration from FaiREE (Li et al., 2022), a post-processing method designed for achieving fairness in finite-sample and distribution-free scenarios, this paper introduces FedFaiREE. The core concept behind FedFaiREE involves distributed utilization of order statistics to conform to fairness constraints and selection of the classifier with the best accuracy (among classifiers that meet the constraints). Specifically, unlike FaiREE, which is restricted to handling i.i.d. centralized data, FedFaiREE is designed to address the challenges presented by decentralized settings. These challenges encompass data decentralization, which incurs limitations such as communication costs associated with updating local data. Additionally, there's the issue of

client heterogeneity. Even though we assume that all training data has been centralized, FaiREE may still encounter bias due to variations among different clients. Finally, client correlation may also violate the assumption of FaiREE. In the context of training decision models across multiple hospitals, it's common for a patient to visit several hospitals, which can influence the assumption of data independence.

Our primary contribution is three-fold: first, we introduce FedFaiREE, a simple yet highly effective approach to ensuring fairness constraints in scenarios with limited samples and distribution-free conditions; second, we provide theoretical guarantees that our method can achieve nearly optimal fairness when the input prediction function is suitable; third, empirically, as demonstrated in Figure 1, we applied existing methods like FairFed (Ezzeldin et al., 2023) and FedAvg (McMahan et al., 2017) with and without FedFaiREE to the Adult dataset (Dua et al., 2017). We found that while existing algorithms are unable to effectively control fairness in real-world applications due to the small sample size in each client, our experiments demonstrate FedFaiREE shows promising performance and can adhere to specific fairness constraints.

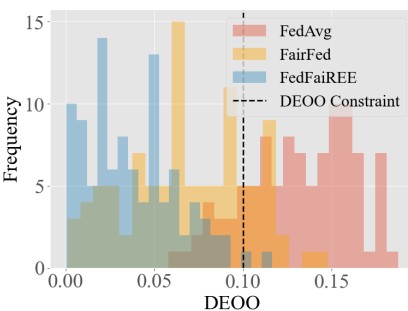

Figure 1: The distribution of DEOO results for FedAvg, both with and without FedFaiREE, comparing to FairFed (Ezzeldin et al., 2023), was examined using the Adult dataset (Dua et al., 2017). See Section 6 for details.

### 1.1 ADDITIONAL RELATED WORK

Existing fairness methodologies in federated Learning predominantly address two key aspects of fairness: fairness among clients and fairness among groups. The former aspect aims to ensure that the global model's performance across individual clients is equitable in terms of equality or contribution (Li et al., 2021; Lyu et al., 2020; Yu et al., 2020; Huang et al., 2020). In contrast, our primary focus in this paper revolves around the latter facet — fairness among groups (Dwork et al., 2012), also referred to as *group fairness*, where the objective is to ensure equitable treatment across different sensitive labels, such as race and gender.

**Existing Group Fairness Techniques.** Conventional approaches can be approximately divided into three categories (Caton & Haas, 2020): pre-processing methods that directly perform debiasing on input data (Zemel et al., 2013; Johndrow & Lum, 2019); in-processing methods that incorporate fairness metrics into model training as part of the objective function (Goh et al., 2016; Cho et al., 2020); post-processing methods that adjust model outputs to enhance fairness (Li et al., 2022; Zeng et al., 2022; Fish et al., 2016).

**Group Fairness Approaches in Federated Learning.** In recent years, there has been a growing amount of work focusing on group fairness in the context of Federated Learning (Ezzeldin et al., 2023; Cui et al., 2021; Zeng et al., 2021; Du et al., 2021; Rodríguez-Gálvez et al., 2021; Chu et al., 2021; Liang et al., 2020; Hu et al., 2022; Papadaki et al., 2022). Most of these studies aim to either introduce fairness principles into the local updates, adapt conventional fairness methods, or perform reweighting during aggregation, or a combination of these strategies. Specifically, Du et al. (2021) proposed AgnosticFair, a framework that utilizes kernel reweighing functions to adjust items in local objective functions, including both loss terms and fairness constraints. Hu et al. (2022) proposed PFFL, which introduces a fairness term into the local objective function and updates the dual parameter $\lambda$ affecting the fairness term as the iterations progress. Zeng et al. (2021) introduced FedFB, a method that adapts Fair Batch, a centralized technique designed to improve fairness among groups by reweighting loss terms for different subgroups, for the FL setting. Ezzeldin et al. (2023) proposed FairFed, an approach that adjusts aggregate weights by considering the disparities between local fairness metrics and the global fairness metric in each training round.

## 2 PRELIMINARIES

In this paper, we address the problem of predicting a binary label, denoted as $Y$, using a set of features, specifically divided into two categories: $X$ and $A$. Here, $X \in \mathcal{X}$ represents non-sensitive features, while $A \in \mathcal{A} = \{0, 1\}$ corresponds to sensitive features. A data point includes $(x, y, a)$,

which correspond to $(X, Y, A)$. For simplicity, we first introduce the concept of *Score-based classifier* (Li et al., 2022).

**Definition 2.1.** *(Score-based classifier) A score-based classifier is an indication function $\hat{Y} = \phi(x, a) = \mathbb{1}\{f(x, a) > c\}$ for a measurable score function $f : \mathcal{X} \times \{0, 1\} \to [0, 1]$ and a constant threshold $c > 0$.*

To assess the fairness of classifier, we introduce two widely-used fairness metrics, Equality of Opportunity and Equalized Odds, which have been extensively utilized in the fairness literature.

**Definition 2.2.** *(Equality of Opportunity (Hardt et al., 2016)) A classifier satisfies Equality of Opportunity if it satisfies the same true positive rate among protected groups: $\mathbb{P}_{X|A=1,Y=1}(\hat{Y} = 1) = \mathbb{P}_{X|A=0,Y=1}(\hat{Y} = 1)$.*

Equality of Opportunity focuses on ensuring an equal opportunity to be predicted as true positive across different groups. However, in practice, achieving strict Equality of Opportunity often is too hard. Therefore, we often introduce a tolerance parameter denoted as $\alpha$ in Equality of Opportunity, as discussed in prior works (Zeng et al., 2022; Li et al., 2022). To be specific, given a classifier $\phi$, the $\alpha$ difference tolerance in Equality of Opportunity can be defined as:

$$|DEOO| \le \alpha, \tag{1}$$

where $DEOO = \mathbb{P}_{X|A=1,Y=1}(\hat{Y} = 1) - \mathbb{P}_{X|A=0,Y=1}(\hat{Y} = 1)$.

**Notation.** To further simplify formulation in the article, we provide notations as follow: $p_a$ signifies the probability of the sensitive attribute $A = a$, i.e., $P(A = a)$. $p_{Y,a}$ represents the probability of label $Y = 1$ given the sensitive attribute $A = a$, i.e., $P(Y = 1 \mid A = a)$, and $q_{Y,a}$ is defined as $1 - p_{Y,a}$. $D$ and $D_i$ represent the datasets for all clients and client $i$, respectively, where $i$ belongs to the set $1, 2, \ldots, S$. $n$ denotes the size of dataset $D$. $T$ represents the ordered scores of elements in dataset $D$. $D_i^{y,a}$ is used to denote the subset of dataset $D_i$ where $Y = y$ and $A = a$. Similar notations apply to $T^{y,a}$ and $n^{y,a}$. $[S]$ denotes the set of integers from 1 to $S$. $\Delta_S$ represents the set of $S$-dimensional vectors $\boldsymbol{v} = (v_1, v_2, \ldots, v_S)$ satisfying the conditions $v_i \ge 0$ and $\sum_{i=1}^{S} v_i = 1$.

## 3 ENABLING FAIR FEDERATED LEARNING

In this section, we introduce FedFaiREE , a **Fed**erated Learning, **Fai**r, distribution-f**REE** algorithm. FedFaiREE has the capability to ensure fairness in scenarios involving finite samples, distribution-free cases and heterogeneity among clients. To incorporate heterogeneity among clients into our model, we make the following assumption.

**Assumption 3.1.** *The training data points within the client $i$ are drawn independently and identically (i.i.d) from distribution $P_i$, while the test data points are sampled from a global distribution that represents a mixture of $P_1, \cdots, P_S$ with weight $\{\pi_i\}_{i \in [S]} \in \Delta_S$. More specifically, we assume that*

$$\left(X_k^i, Y_k^i\right) \sim P_i, \ (X^{test}, Y^{test}) \sim P^{mix} = \sum_{i=1}^{S} \pi_i P_i$$

This implies that each client $i$ has its own distribution $P_i$, and test data points are randomly sampled from client $i$ with a probability of $\pi_i$.

### 3.1 PROBLEM FORMULATION AND FEDFAIREE OVERVIEW

Suppose we have $S$ clients and a pre-trained score-based classifier $\phi_0(x, a) = \mathbf{1}\{f(x, a) > c\}$, each with their own local data $D_i = D_i^{0,0} \cup D_i^{0,1} \cup D_i^{1,0} \cup D_i^{1,1}$. Here, $i \in [S]$ represents each client, and $D_i^{y,a} = \{x_{i,1}^{y,a}, x_{i,2}^{y,a}, \cdots, x_{i,n^{y,a}}^{y,a}\}$ is a subset of data points in $D_i$, where $Y = y \in \{0, 1\}$ and $A = a \in \{0, 1\}$ denote the label and sensitive attribute, respectively. Our objective is to determine approximate thresholds $\lambda_0$ and $\lambda_1$ for $A = 0$ and $A = 1$ to construct the corresponding classifier $\phi(x, a) = \mathbf{1}\{f(x, a) > \lambda_a\}$ that yields the optimal misclassification performance while adhering to specific fairness constraints.

To achieve this, inspired by FaiREE (Li et al., 2022), we primarily leverage the rank of scores in the training set, capitalizing on certain properties of order statistics. Figure 2 provides an overview of our algorithm, which mainly comprises three key components:

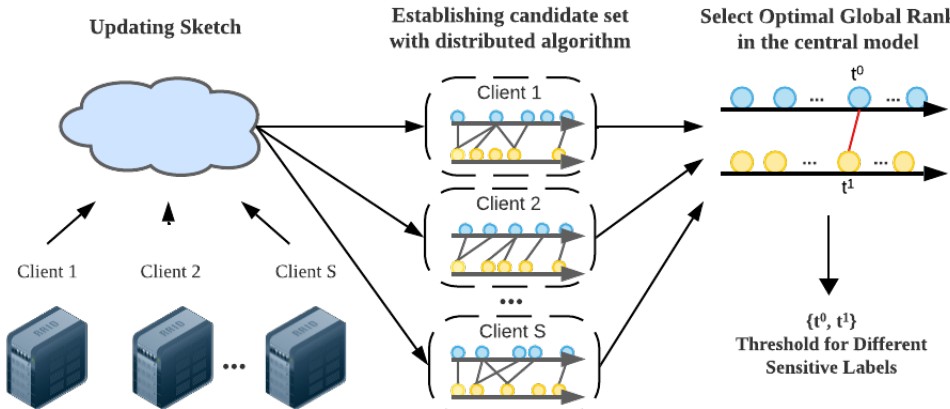

Figure 2: **Overview of FedFaiREE.** With S clients and a pre-trained model in consideration, each ball in the image symbolizes a score in the training set. The color of the balls represents different sensitive labels, while the gray edges depict local ranks of threshold pairs (each global classifier's threshold pair corresponds to S local ranks). Notably, the red edge signifies the chosen global classifier with thresholds $t^0, t^1$ for sensitive labels $A = 0$ and $A = 1$, respectively.

**Step 1: Update local scores.** For each local client $s$, we apply the given scorer $f$ to obtain scores $t_{i,j}^{y,a} = f(x_{i,j}^{y,a})$. Subsequently, we sort score set $T_i^{y,a} = \{t_{i,1}^{y,a}, t_{i,2}^{y,a}, \cdots, t_{i,n_i^{y,a}}^{y,a}\}$ in the order of non-decreasing . To minimize communication costs, we update a sketch of $T_i^{y,a}$.

**Step 2: Construct the candidate set with distributed quantile algorithm.** Incorporating specific fairness constraints and accounting for client heterogeneity, we utilize the updated sketches to construct a candidate set comprising rank pairs $(\mathbf{k}^{1,0}, \mathbf{k}^{1,1})$. This construction involves both order statistics and distributed quantile algorithms. Here, $\mathbf{k}^{1,a}$ represents the $S$ local ranks as defined in Section 3.2. Each pair represents a classifier with $\lambda_a$ chosen as the estimated global $k^{1,a}$-th value in the sorted global score set $T^{1,a}$, i.e., $\hat{t}_{(k^{1,a})}^{1,a}$, where $k^{1,a}$ is the corresponding global rank of $\mathbf{k}^{1,a}$.

**Step 3: Select the best threshold in the central model.** To select the optimal rank pair from among multiple classifiers, we choose the rank pair that minimizes the estimated misclassification error, all while accounting for client heterogeneity. Consequently, the resulting classifier can be represented as $\hat{\phi}(x, a) = \mathbf{1}\{f(x, a) > \hat{t}_{(k^{1,a})}^{1,a}\}$.

In the subsequent section, we illustrate our approach using Equality of Opportunity as our targeted group fairness constraint, providing a detailed description of our algorithm. Additionally, it's important to emphasize that, like FaiREE, FedFaiREE is adaptable to various fairness notions, with the added capability of accommodating even more diverse situations. We would discuss more fairness concepts like Equalized Odds and label shift scenario in Sections 5.

## 3.2 ESTABLISHING CANDIDATE SET WITH DISTRIBUTED QUANTILE ALGORITHM

To select rank pairs that satisfy fairness constraints, we leverage the properties of order statistics. Specifically, we consider that $k^{1,a}$ represents the rank in the sorted $T^{1,a}$. However, unlike FaiREE, we need to consider heterogeneity among clients, and further define $k_i^{1,a}$ to represent the corresponding rank of $t_{(k^{1,a})}^{1,a}$ in the sorted $T_i^{1,a}$, where $i \in [S]$ and $k_i^{1,a}$ satisfies $t_{i,(k_i^{1,a})}^{1,a} \leq t_{(k^{1,a})}^{1,a} < t_{i,(k_i^{1,a}+1)}^{1,a}$. Using this approach, we make an observation in controlling fairness.

**Proposition 3.1.** *Under Assumption 3.1, for $a \in \{0,1\}$, consider $k^{1,a} \in \{1, \ldots, n^{1,a}\}$, the corresponding $k_i^{1,a}$ for $i \in [S]$ and the score-based classifier $\phi(x, a) = \mathbb{1}\{f(x, a) > t_{(k^{1,a})}^{1,a}\}$. Define*

$$h_{y,a}(\mathbf{u}, \mathbf{v}) = \mathbb{P}\left(\sum_{i=1}^{S} \pi_i^{y,a} Q\left(u_i, n_i^{y,a} + 1 - u_i\right) - \sum_{i=1}^{S} \pi_i^{y,1-a} Q\left(v_i, n_i^{y,1-a} + 1 - v_i\right) \geq \alpha\right).$$

*Then we have:*

$$\mathbb{P}(|DEOO(\phi)| > \alpha) \leq h_{1,0}(\mathbf{k}^{1,0} + \mathbf{1}, \mathbf{k}^{1,1}) + h_{1,1}(\mathbf{k}^{1,1} + \mathbf{1}, \mathbf{k}^{1,0}) \tag{2}$$

*Where $\boldsymbol{k}^{1,a} = (k_1^{1,a}, \cdots, k_S^{1,a})$, $\pi_i^{1,a} = \mathbb{P}(sampling\ x\ from\ client\ i\ |\ sampling\ x\ with\ label\ Y = 1\ and\ A = a)$, and $Q(\alpha, \beta)$ are independent random variables and $Q(\alpha, \beta) \sim Beta(\alpha, \beta)$.*

This proposition enables us to select classifiers that satisfy fairness constraints with arbitrary finite sample and no distributional assumption. Moreover, $Q(\alpha, \beta)$ can be efficiently estimated by Monte Carlo simulations in applications. Specifically, we approximated $Q(\alpha, \beta)$ by conducting random sampling 1000 times in our experiment, yielding a highly satisfactory approximation.

Due to the need of computing local ranks to make use of Proposition 3.1, it is important to consider the tradeoff between accuracy and communication cost in real applications. In particular, we can adopt distributed quantile algorithms to reduce communication costs while controlling errors in calculating local ranks. Therefore, we present an alternative formulation of Proposition 3.1 to allow errors in the local rank calculation. To begin with, we introduce the concept of approximate quantiles and ranks (Luo et al., 2016; Lu et al., 2023).

**Definition 3.2.** *($\varepsilon$-approximate $\beta$-quantile and rank of a given set) For an error $\varepsilon \in (0, 1)$, the $\varepsilon$-approximate $\beta$-quantile of a given set is any element with rank between $(\beta - \varepsilon)N$ and $(\beta + \varepsilon)N$, where $N$ is the total number of elements in set. Further, the $\varepsilon$-approximate rank of a element in a given set is any rank between $(\beta - \varepsilon)N$ and $(\beta + \varepsilon)N$ where $\beta N$ represents the real rank.*

Under Definition 3.2, if the rank estimation method produces $\varepsilon$-approximate ranks, it is possible to correspondingly modify Proposition 3.1.

**Proposition 3.2.** *Under Assumption 3.1, for $a \in \{0, 1\}$, consider $k^{1,a} \in \{1, \ldots, n^{1,a}\}$, the corresponding $\hat{k}_i^{1,a}$ for $i \in [S]$ which are $\varepsilon$-approximate ranks and the score-based classifier $\phi(x, a) = \mathbb{1}\{f(x, a) > t_{(k^{1,a})}^{1,a}\}$. Define*

$$h_{y,a}(\boldsymbol{u}, \boldsymbol{v}) = \mathbb{P}\left(\sum_{i=1}^{S} \pi_i^{y,a} Q\left(u_i, n_i^{y,a} + 1 - u_i\right) - \sum_{i=1}^{S} \pi_i^{y,1-a} Q\left(v_i, n_i^{y,1-a} + 1 - v_i\right) \geq \alpha\right).$$

*Then we have:*

$$\mathbb{P}(|DEOO(\phi)| > \alpha) \leq h_{1,0}(\boldsymbol{M}^{1,0}, \boldsymbol{m}^{1,1}) + h_{1,1}(\boldsymbol{M}^{1,1}, \boldsymbol{m}^{1,0}), \tag{3}$$

*where $\pi_i^{1,a}$ is defined in Proposition 3.1, $\boldsymbol{M}^{1,a} = (M_1^{1,a}, \cdots, M_S^{1,a})$, $\boldsymbol{m}^{1,a} = (m_1^{1,a}, \cdots, m_S^{1,a})$, $M_i^{1,a} = max(\lceil \hat{k}_i^{1,a} + \varepsilon n_i^{1,a} \rceil, n_i^{1,a} + 1)$, $m_i^{1,a} = min(\lceil \hat{k}_i^{1,a} - \varepsilon n_i^{1,a} \rceil, 0)$, and $Q(\alpha, \beta)$ are independent random variables and $Q(\alpha, \beta) \sim Beta(\alpha, \beta)$. Especially, we define $Q(0, \beta) = 0$ and $Q(\alpha, 0) = 1$ for $\alpha, \beta \neq 0$.*

In practical distributed settings, calculating the exact local rank in Proposition 3.2 is generally hard due to communication constraints. By adopting approximate $\varepsilon$ and related parameters in distributed quantile algorithm, we strike a balance between accuracy and communication cost, enabling the effective implementation of our algorithm in distributed environments.

In our experiments, we implemented the Q-digest (Shrivastava et al., 2004), a tree-based sketching distributed quantile algorithm commonly used for efficiently approximating quantiles and ranks computation with rigorous theory controlling the error. Due to the inherent characteristics of the Q-digest algorithm, it only yields approximate quantiles and ranks that tend to be greater than their true values. However, considering the adaptability of other distributed quantile algorithms and aiming to reduce the absolute value of $\varepsilon$, we take into account both upward and downward estimation deviations as described in Definition 3.2.

Based on Proposition 3.2, we can construct the candidate set $K$ as

$$K = \{(\boldsymbol{k}^{1,0}, \boldsymbol{k}^{1,1}) | L(\boldsymbol{k}^{1,0}, \boldsymbol{k}^{1,1}) < 1 - \beta\}, \tag{4}$$

where $\boldsymbol{k}^{1,a} = (\hat{k}_1^{1,a}, \cdots, \hat{k}_S^{1,a})$, and $L(\boldsymbol{k}^{1,0}, \boldsymbol{k}^{1,1})$ represents the right-hand side of Inequality 3.

### 3.3 SELECTION FOR THE OPTIMAL THRESHOLD

In this subsection, we elaborate our method for selecting the optimal threshold. For a given pair $(k^{1,0}, k^{1,1})$ from the candidate set, we exploit the properties of order statistics to compute estimated misclassification error and then select the pair minimizing the estimated error.

---

**Algorithm 1** FedFaiREE for DEOO

---

**Input:** Train dataset $D_i = D_i^{0,0} \cup D_i^{0,1} \cup D_i^{1,0} \cup D_i^{1,1}$; pre-trained classifier $\phi_0$ with function f; fairness constraint parameter $\alpha$ ; Confidence level parameter $\beta$; Weights of different clients $\pi$

**Output:** classifier $\hat{\phi}(x,a) = \mathbf{1}\{f(x,a) > t_{(k^{1,a})}^{1,a}\}$

**Client Side:**

**for** i=1,2,...,S **do**

     Score on train data points in $D_i$ and get $T_i^{y,a} = \{t_{i,1}^{y,a}, t_{i,2}^{y,a}, \cdots, t_{i,n_i^{y,a}}^{y,a}\}$

     Sort $T_i^{y,a}$ and calculate q-digest of $T_i^{y,a}$ on client $i$

     Update digest to server

**end for**

**Server Side:**

Construct $K$ by $K = \{(\boldsymbol{k}^{1,0}, \boldsymbol{k}^{1,1}) | L(\boldsymbol{k}^{1,0}, \boldsymbol{k}^{1,1}) < 1 - \beta\}$

Select optimal $(\boldsymbol{k}_0, \boldsymbol{k}_1)$ by minimizing Equation 5 using estimated values $\hat{p}_a^i$, $\hat{p}_{Y,a}^i$ and $\hat{q}_{Y,a}^i$

---

To facilitate this, we need to compute the approximate ranks of $t_{(k^{1,0})}^{1,0}$ and $t_{(k^{1,1})}^{1,1}$ in the sorted sets $T_i^{0,0}$ and $T_i^{0,1}$, where $i \in [S]$, respectively. Specifically, we determine $k_i^{0,a}$ such that $t_{i,(k_i^{0,a})}^{0,a} \leq t_{(k^{1,a})}^{1,a} < t_{i,(k_i^{0,a}+1)}^{0,a}$ for $a \in \{0,1\}$. To simplify, in following sections, we assume the corresponding $\hat{k}_i^{1,a}$ for $i \in [S]$ are $\varepsilon$-approximate ranks and the estimated quantiles presented by distributed quantile algorithm are $\varepsilon$-approximate quantiles. Then, we commence by presenting our observation on the estimation of misclassification error through the following proposition.

**Proposition 3.3.** *Under Assumption 3.1, the misclassification error can be estimated by*

$$
\begin{aligned}
\hat{\mathbb{P}}\left(\hat{\phi}(x,a) \neq Y\right) = \sum_{i=1}^{S} \pi_i \Big[ & \frac{\hat{k}_i^{1,0} + 0.5}{n_i^{1,0} + 1} p_0^i p_{Y,0}^i + \frac{\hat{k}_i^{1,1} + 0.5}{n_i^{1,1} + 1} p_1^i p_{Y,1}^i + \frac{n_i^{0,0} + 0.5 - \hat{k}_i^{0,0}}{n_i^{0,0} + 1} p_0^i q_{Y,0}^i \\
& + \frac{n_i^{0,1} + 0.5 - \hat{k}_i^{0,1}}{n_i^{0,1} + 1} p_1^i q_{Y,1}^i \Big]
\end{aligned}
\tag{5}
$$

*Further, the discrepancy between empirical error and true error is upper bounded by the following:*

$$
\left| \mathbb{P}\left(\hat{\phi}(x,a) \neq Y\right) - \hat{\mathbb{P}}\left(\hat{\phi}(x,a) \neq Y\right) \right| \leq \theta,
\tag{6}
$$

*where* $\theta = \sum_{i=1}^{S} \pi_i \left[ e_i^{0,0} p_0^i q_{Y,0}^i + e_i^{0,1} p_0^i p_{Y,0}^i + e_i^{1,0} p_1^i q_{Y,1}^i + e_i^{1,1} p_1^i p_{Y,1}^i \right]$, $e_i^{y,a} = \frac{2\lfloor \varepsilon n_i^{y,a} \rfloor + 1}{2(n_i^{y,a}+1)}$

Proposition 3.3 provides a method for estimating the overall misclassification error using data from the training set with Equation 5. However, we may not have exact knowledge of the probabilities $p_a^i$ and $p_{Y,a}^i$. In such cases, we can use the estimated values $\hat{p}_a^i = \frac{n_i^{0,a} + n_i^{1,a}}{n_i^{0,0} + n_i^{0,1} + n_i^{1,0} + n_i^{1,1}}$, $\hat{p}_{Y,a}^i = \frac{n_i^{1,a}}{n_i^{0,a} + n_i^{1,a}}$, $\hat{q}_{Y,a}^i = 1 - \hat{p}_{Y,a}^i$ to calculate the empirical error. We will further present a theorem to show that we can achieve a desirable accuracy using the estimated values in Section 4.

At the end of this section, we provide a concise summary of our algorithm in Algorithm 1. It's worth noting that while in our experiment, we assume that $\pi_i$ is proportional to $n_i$, we may not know the exact values of $\pi_i$ in real applications. To enhance the robustness of our approach in such real-world scenarios, one can consider introducing a hypothesis space denoted as $H(\pi)$ to model the range of $\pi$ and incorporate $\max_{\pi \in H(\pi)}$ into equations 4 and 5.

## 4 THEORETICAL GUARANTEES

In this section, we provide the accuracy analysis for FedFaiREE. To mitigate situations where there might be extreme initial pre-trained classifier, we introduce the following assumption.

**Assumption 4.1.** *The distribution of $f(x,a)$ exhibits the following property. When conducting $N$ independent samplings to form a sample set, let $q_0$ be the $\beta$-quantile of the sample set. There exist function $\delta : \mathbb{N} \rightarrow \mathbb{R}$, constant $\gamma > 0$, such that $\lim_{N \to \infty} \delta(N) = 0$ and with a probability of at least $1 - \delta(N)$, for any $q$ considered as an $\varepsilon$-approximate $\beta$-quantile of the sample set, it satisfies that , $q$ lies within the $\gamma\varepsilon$-neighborhood of $q_0$.*

In simpler terms, with Assumption 1, we can avoid significant deviations between our approximated quantile and the actual quantile in certain extreme cases. Moreover, in the following theorem, we establish a theoretical basis for the accuracy of FedFaiREE. To facilitate accurate comparisons, we introduce the notion of the "fair Bayes-optimal classifier", a concept developed by Zeng et al. (2022) meaning Bayes-optimal fair classification under specific "level of disparity". The precise definition of the fair Bayes-optimal classifier under DEOO can be found in Lemma A.2. To be concise, we denotes the standard Bayes-optimal classifier without fairness constraint as $\phi^*(x, a) = \mathbb{1}\{f^*(x, a) > 1/2\}$, where $f^* \in \arg\min_f[\mathbb{P}(Y \neq \mathbb{1}\{f(x, a) > 1/2\})]$.

**Theorem 4.2.** *Under Assumption 3.1 and 4.1, given $\alpha' < \alpha$. Suppose $\hat{\phi}$ is the final output of FedFaiREE, we have:*

*(1) $|DEOO(\hat{\phi})| < \alpha$ with probability $(1 - \delta)^N$, where $N$ is the size of the candidate set.*

*(2) Suppose the density distribution functions of $f^*$ under $A = a, Y = 1$ are continuous. When the input classifier $f$ satisfies $|f(x, a) - f^*(x, a)| \leq \epsilon_0$, for any $\epsilon > 0$ such that $F^*_{(+)}(\epsilon + \gamma\varepsilon) \leq \frac{\alpha - \alpha'}{2} - F^*_{(+)}(2\epsilon_0)$, we have*

$$\mathbb{P}(\hat{\phi}(x, a) \neq Y) - \mathbb{P}(\phi^*_{\alpha'}(x, a) \neq Y) \leq 2F^*_{(+)}(2\epsilon_0) + 2F^*_{(+)}(\epsilon + \gamma\varepsilon) + 8\epsilon^2 + 20\epsilon + 2\theta \quad (7)$$

*with probability $1 - 4\sum_{i=1}^{S}(e^{-2n_i^{0,0}\epsilon^2} + e^{-2n_i^{0,1}\epsilon^2}) - \sum_{a=0}^{1}\prod_{i=1}^{S}\left(1 - F_{i(-)}^{1,a}(2\epsilon)\right)^{n_i^{1,a}} - \delta$, where $\delta = \delta^{1,0}(n^{1,0}) + \delta^{1,1}(n^{1,1})$, $\theta$ is defined in Proposition 3.3 and the definition of $F_{(+)}$ and $F_{(-)}$ are shown in Lemma A.4*

This theorem provides assurance that our method can achieve almost the optimal misclassification error with DEOO constraints, provided that the input classifier is chosen appropriately i.e. is close enough to the Bayes-optimal one. This theorem underscores the effectiveness of our approach in minimizing errors when ensuring fairness in a distribution-free and finite-sample manner.

# 5 APPLICATIONS IN DIFFERENT SCENARIOS

## 5.1 LABEL SHIFT IN TEST SET

In this section, we explore the application of our algorithm in various scenarios. First, we assume the presence of a label shift in the test set, a situation that frequently encountered in real-world applications (Plassier et al., 2023; Tian et al., 2023). To do so, we first need to revise Assumption 3.1 to adapt extension settings. Specifically, we introduce the following assumption.

**Assumption 5.1.** *The training data points on client $i$ are i.i.d drawn from the distribution $P_i$, and we further assume the global distribution $P$ is mixture of $P_1, \cdots, P_S$ with weight $\{\pi_i\}_{i\in[S]} \in \Delta_S$, while the test data points are sampled from another distribution $P_i$, heterogeneity between $P$ and which induced due to label shift, that is, we assume that*

$$\begin{aligned}
\left(X_k^i, Y_k^i\right) \sim P_i, P^{mix} = \sum_{i=1}^{S} \pi_i P_i = P(X, A|Y) * P^{mix}(Y), \\
\left(X^{test}, Y^{test}\right) \sim P_i = P(X, A|Y) * P_i(Y)
\end{aligned} \quad (8)$$

We note that FedFaiREE can be adapted to Assumption 5.1 by modifying the target function for the optimal rank selection from Equation 5 to the following equation:

$$\begin{aligned}
\hat{\mathbb{P}}\left(\hat{\phi}(x, a) \neq Y\right) = \sum_{i=1}^{S} \pi_i \big[ & \frac{\hat{k}_i^{1,0} + 0.5}{n_i^{1,0} + 1} p_0^i p_{Y,0}^i w^{1,0} + \frac{\hat{k}_i^{1,1} + 0.5}{n_i^{1,1} + 1} p_1^i p_{Y,1}^i w^{1,1} \\
& + \frac{n_i^{0,0} + 0.5 - \hat{k}_i^{0,0}}{n_i^{0,0} + 1} p_0^i q_{Y,0}^i w^{0,0} + \frac{n_i^{0,1} + 0.5 - \hat{k}_i^{0,1}}{n_i^{0,1} + 1} p_1^i q_{Y,1}^i w^{0,1} \big],
\end{aligned} \quad (9)$$

where $w^{y,a} = \frac{p_a^{S+1} p_{Y,a}^{S+1}}{p_a p_{Y,a}}$. In Appendix A.4, we provide a detailed proposition to ensure the accuracy of our estimations and present a concise algorithm. Furthermore, to account for label shift scenarios, we offer a theorem guarantee as a revised version of 4.2 at the end of this subsection.

**Theorem 5.2.** *Under Assumption 4.1 and 5.1, given $\alpha' < \alpha$. Suppose $\hat{\phi}$ is the final output of FedFaiREE, we have:*

*(1) $|DEOO(\hat{\phi})| < \alpha$ with probability $(1 - \delta)^N$, where $N$ is the size of the candidate set.*

*(2) Suppose the density distribution functions of $f^*$ under $A = a, Y = 1$ are continuous. When the input classifier $f$ satisfies $|f(x, a) - f^*(x, a)| \leq \epsilon_0$, for any $\epsilon > 0$ such that $F^*_{(+)}(\epsilon + \gamma\varepsilon) \leq \frac{\alpha - \alpha'}{2} - F^*_{(+)}(2\epsilon_0)$, we have*

$$\mathbb{P}(\hat{\phi}(x, a) \neq Y) - \mathbb{P}(\phi^*_{\alpha'}(x, a) \neq Y) \leq 2F^*_{(+)}(2\epsilon_0) + 2F^*_{(+)}(\epsilon + \gamma\varepsilon) + 2\theta' + O(\epsilon) \tag{10}$$

*with probability $1 - 4\sum_{i=1}^{S}(e^{-2n_i^{0,0}\epsilon^2} + e^{-2n_i^{0,1}\epsilon^2}) - \sum_{a=0}^{1}\prod_{i=1}^{S}\left(1 - F_{i(-)}^{1,a}(2\epsilon)\right)^{n_i^{1,a}} - \delta$, where the definitions of $\delta$, $F_{(+)}$, $F_{(-)}$ are same with Theorem 4.2, $\theta'$ is defined in Proposition A.1.*

In summary, Theorem 5.2 assures that our FedFaiREE algorithm can effectively control fairness and maintain accuracy in situations where label shift is present in the test data. These guarantees are essential for deploying fair and accurate machine learning models in practical applications.

## 5.2 EQUALIZED ODDS

We have also explored the potential extension of our algorithm to fairness indicators beyond DEOO. Specifically, in this subsection, we will discuss its application to Equalized Odds. Applications on more fairness notions are presented in Appendix B.

**Definition 5.3.** *(Equalized Odds (Hardt et al., 2016)) A classifier satisfies Equalized Odds if it satisfies the following equality: $\mathbb{P}_{X|A=1,Y=1}(\widehat{Y} = 1) = \mathbb{P}_{X|A=0,Y=1}(\widehat{Y} = 1)$ and $\mathbb{P}_{X|A=1,Y=0}(\widehat{Y} = 1) = \mathbb{P}_{X|A=0,Y=0}(\widehat{Y} = 1)$.*

Similarly, we can express the fairness constraints under Equalized Odds as $|DEO| \preceq (\alpha_1, \alpha_2)$, which is equivalent to $|\mathbb{P}_{X|A=1,Y=1}(\widehat{Y} = 1) - \mathbb{P}_{X|A=0,Y=1}(\widehat{Y} = 1)| \leq \alpha_1$ and $|\mathbb{P}_{X|A=1,Y=0}(\widehat{Y} = 1) - \mathbb{P}_{X|A=0,Y=0}(\widehat{Y} = 1)| \leq \alpha_2$. Hence, in order to consider two fairness constraints simultaneously, we modify Equation 4 as follow.

$$K = \{(\boldsymbol{k}^{*,0}, \boldsymbol{k}^{*,1})|L(\boldsymbol{k}^{*,0}, \boldsymbol{k}^{*,1}) = h^*_{1,1} + h^*_{1,0} + h^*_{0,1} + h^*_{0,0} < 1 - \beta\}, \tag{11}$$

where $\boldsymbol{k}^{*,a} = (\hat{k}_1^{0,a}, \cdots, \hat{k}_S^{0,a}, \hat{k}_1^{1,a}, \cdots, \hat{k}_S^{1,a})$ and $h^*_{y,a}$ are function of $\boldsymbol{k}^{*,a}$ defined in Proposition A.2. Additional details, propositions can be found in Appendix A.5. This equation allows us to construct a candidate set under DEO fairness constraints, enabling us to apply our algorithm to achieve Equalized Odds. Furthermore, we provide theoretical guarantees for DEO fairness.

**Theorem 5.4.** *Under Assumption 3.1 and 4.1, given $\alpha' < \alpha$. Suppose $\hat{\phi}$ is the final output of FedFaiREE with target DEO constraint, we have:*

*(1) $|DEO(\hat{\phi})| < \alpha$ with probability $(1 - \delta)^N$, where $N$ is the size of the candidate set.*

*(2) Suppose the density distribution functions of $f^*$ under $A = a, Y = 1$ are continuous. When the input classifier $f$ satisfies $|f(x, a) - f^*(x, a)| \leq \epsilon_0$, for any $\epsilon > 0$ such that $F^*_{(+)}(\epsilon + \gamma\varepsilon) \leq \frac{\alpha - \alpha'}{2} - F^*_{(+)}(2\epsilon_0)$, we have*

$$\mathbb{P}(\hat{\phi}(x, a) \neq Y) - \mathbb{P}(\phi^*_{\alpha'}(x, a) \neq Y) \leq 2F^*_{(+)}(2\epsilon_0) + 2F^*_{(+)}(\epsilon + \gamma\varepsilon) + 2\theta + O(\epsilon) \tag{12}$$

*with probability $1 - 4\sum_{i=1}^{S}(e^{-2n_i^{0,0}\epsilon^2} + e^{-2n_i^{0,1}\epsilon^2}) - \sum_{a=0}^{1}\prod_{i=1}^{S}\left(1 - F_{i(-)}^{1,a}(2\epsilon)\right)^{n_i^{1,a}} - \delta$, where the definitions of $\delta$, $\theta$, $F_{(+)}$, $F_{(-)}$ are same with Theorem 4.2.*

## 6 EXPERIMENT

In this section, we study the performance of FedFaiREE on real datasets including Adult(Dua et al., 2017) and Compas(Dieterich et al., 2016). In particular, we employed FedFaiREE on FedAvg(McMahan et al., 2017), AFL(Mohri et al., 2019), FedFB(Zeng et al., 2021), FairFed(Ezzeldin

Table 1: **Results on Adult and Compas dataset.** We conducted 100 experimental repetitions for each model on both datasets and compared the accuracy and fairness indicators of different models. The FedFaiREE and $\alpha$ columns indicate whether FedFaiREE was used or not and the fairness constraint. Confidence level $\beta$ is set to be 95% throughout the experiments. $\overline{ACC}$ and $\overline{|DEOO|}$ represent the averages of accuracy and DEOO (defined in Equation 1). $|DEOO|_{95}$ represents the 95% quantile of DEOO since we set the confidence level of FedFaiREE to 95% in our experiments.

| Model | FedFaiREE | Adult | | | | Compas | | | |
| --- | --- | --- | --- | --- | --- | --- | --- | --- | --- |
| | | $\alpha$ | $\overline{ACC}$ | $\overline{|DEOO|}$ | $|DEOO|_{95}$ | $\alpha$ | $\overline{ACC}$ | $\overline{|DEOO|}$ | $|DEOO|_{95}$ |
| **FedAvg** | ✗ | / | 0.844 | 0.131 | 0.178 | / | 0.662 | 0.126 | 0.223 |
| | ✓ | 0.10 | 0.843 | **0.038** | **0.083** | 0.15 | 0.659 | **0.051** | **0.137** |
| **AFL** | ✗ | / | 0.848 | 0.101 | 0.169 | / | 0.643 | 0.097 | 0.170 |
| | ✓ | 0.10 | 0.848 | **0.034** | **0.081** | 0.15 | 0.641 | **0.051** | **0.108** |
| **FedFB** | ✗ | / | 0.850 | 0.057 | 0.117 | / | 0.642 | 0.107 | 0.174 |
| | ✓ | 0.10 | 0.850 | **0.036** | **0.083** | 0.15 | 0.641 | **0.062** | **0.125** |
| **FairFed** | ✗ | / | 0.842 | 0.069 | 0.118 | / | 0.648 | 0.097 | 0.166 |
| | ✓ | 0.10 | 0.841 | **0.037** | **0.081** | 0.15 | 0.645 | **0.047** | **0.114** |

et al., 2023) and training all algorithm using a two layers neural networks. See Appendix C for further details of experiments, including hyperparameter range, detailed model information and more.

**Dataset.** Adult dataset (Dua et al., 2017), which is employed for the prediction task that determine whether an individual's income exceeds $50,000, comprises 45,222 samples, featuring various attributes including age, education, and more. The sensitive feature in our analysis is gender. Compas dataset (Dieterich et al., 2016), whose task is to predict whether a person will conduct crime in the future, comprises 7214 samples. The sensitive feature is gender.

**Data Processing.** To replicate the decentralized conditions and account for heterogeneity across clients, we adopted the approach introduced by Ezzeldin et al. (2023). Specifically, we initiated the process by randomly sampling proportions for various sensitive attributes within each client, using the Dirichlet distribution. Subsequently, we partitioned the dataset into client-specific subsets based on these proportions. Within each of these subsets, we performed an 80-20 split, allocating 80% of the data as the local client training set and reserving the remaining 20% for the test set. For the numerical experiments, we repeated this procedure 100 times on both Adult and Compas datasets.

**Result and Analysis.** Table 1 presents the results obtained from experiments conducted on both the Adult and Compas datasets. These results showcase that FedFaiREE achieved desirable performance across both datasets. The column labeled FedFaiREE indicates whether FedFaiREE was used and the columns labeled $\alpha$ specify the fairness constraint. Our findings demonstrate that FedFaiREE, with its unique, distribution-free approach to fairness constraints under finite-sample, consistently outperforms the original models in controlling DEOO while maintaining relatively high accuracy. It is worth noting that FedFaiREE achieves significant performance improvements even when applied to FedAvg, the most fundamental model. This indicates the wide applicability and potential of FedFaiREE across various settings. Notably, FedFaiREE was employed with a confidence level of $\beta = 0.95$ throughout the experiments, and it successfully controlled the 95th percentile of DEOO, showcasing its robustness.

## 7 CONCLUSION

In this paper, we introduce FedFaiREE, a novel, distribution-free approach aimed at guaranteeing fairness constraints under the federated learning setting. The unique strength of FedFaiREE lies in its ability to address real-world concerns of federated learning, such as client heterogeneity, small samples and communication costs. We showcase the adaptability of FedFaiREE by demonstrating its applicability to a wide range of group fairness notions and various scenarios such as label shifts. Our experiments provide further validation for its practical value. For future works, there are several promising directions. First, extending FedFaiREE to tasks that go beyond binary label prediction could open up new avenues for practical applications. Additionally, exploring more efficient distributed quantile algorithms for rank and quantile calculations within the FedFaiREE framework could further improve its scalability and performance.

## REPRODUCIBILITY STATEMENT

The code and dataset for our work can be found in the supplemental materials. To ensure reproducibility, we would like to note that we set random seeds in the range of 0 to 99 for our experiments on Compas dataset. Given that our splitting method allows for potential heterogeneity and varying dataset sizes, which might result in empty datasets, performing "split failed" in our code, we used random seeds in the range of 0 to 111 for the adult dataset when the parameter for the Dirichlet distribution was set to 1. For specific hyperparameter selections, please refer to Table 2.

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

# A  PROOFS

## A.1  PROOF FOR PROPOSITION 3.1 AND 3.2

We first introduce following lemma

**Lemma A.1.** *If $t_i^{y,a}$ is variable with continuous density function, we have*

$$F_i^{y,a}\left(t_{i,\left(k_i^{y,a}\right)}^{y,a}\right) \sim \text{Beta}\left(k_i^{y,a}, n_i^{y,a} - k_i^{y,a} + 1\right)$$

.

*Proof of Lemma A.1.* $F_i^{y,a}$ represents the continuous cumulative distribution functions of $t_i^{y,a}$, and thus we have $F_i^{y,a}(t_i^{y,a}) \sim U(0,1)$. Furthermore, as $F_i^{y,a}\left(t_{i,\left(k_i^{y,a}\right)}^{y,a}\right)$ denotes the $k_i^{y,a}$-th order statistic of $n_i^{y,a}$ i.i.d samples from $U(0,1)$, we can conclude that $F^{y,a}\left(t_{i,\left(k_i^{y,a}\right)}^{y,a}\right) \sim$ Beta$\left(k_i^{y,a}, n^{y,a} - k_i^{y,a} + 1\right)$ □

Back to proof of the Proposition 3.1, the classifier is

*Proof of Proposition 3.1.*

$$\phi = \begin{cases} \mathbb{1}\left\{f(x,0) > t_{(k^{1,0})}^{1,0}\right\}, a = 0 \\ \mathbb{1}\left\{f(x,1) > t_{(k^{1,1})}^{1,1}\right\}, a = 1 \end{cases}$$

we have:

$$\begin{aligned}
\mathbb{P}(|DEOO(\phi)| > \alpha) &= \mathbb{P}\left(\left|F^{1,1}\left(t_{(k^{1,1})}^{1,1}\right) - F^{1,0}\left(t_{(k^{1,0})}^{1,0}\right)\right| > \alpha\right) \\
&= \mathbb{P}\left(\sum_{i=1}^{S}\pi_i^{1,1}F_i^{1,1}\left(t_{(k^{1,1})}^{1,1}\right) - \sum_{i=1}^{S}\pi_i^{1,0}F_i^{1,0}\left(t_{(k^{1,0})}^{1,0}\right) > \alpha\right) \\
&\quad + \mathbb{P}\left(\sum_{i=1}^{S}\pi_i^{1,1}F_i^{1,1}\left(t_{(k^{1,1})}^{1,1}\right) - \sum_{i=1}^{S}\pi_i^{1,0}F_i^{1,0}\left(t_{(k^{1,0})}^{1,0}\right) < -\alpha\right) \\
&\triangleq A + B
\end{aligned}$$

So we only need to calculate $A$ and $B$ and It is easy to prove that we only need to consider the continuous density function case..

$$\begin{aligned}
A &= \mathbb{P}\left(\sum_{i=1}^{S}\pi_i^{1,1}F_i^{1,1}\left(t_{(k^{1,1})}^{1,1}\right) - \sum_{i=1}^{S}\pi_i^{1,0}F_i^{1,0}\left(t_{(k^{1,0})}^{1,0}\right) > \alpha\right) \\
&\leq \mathbb{P}\left(\sum_{i=1}^{S}\pi_i^{1,1}F_i^{1,1}\left(t_{i,\left(k_i^{1,1}+1\right)}^{1,1}\right) - \sum_{i=1}^{S}\pi_i^{1,0}F_i^{1,0}\left(t_{i,\left(k_i^{1,0}\right)}^{1,0}\right) > \alpha\right)
\end{aligned}$$

Considering lemma A.1 and similar result for B, we complete the proof. □

For the proof of Proposition 3.2, we can adjust the estimation of A by introducing the error generated in rank calculation. Specifically, we show that

*Sketch proof of Proposition 3.2.*

$$\begin{aligned}
A &= \mathbb{P}\left(\sum_{i=1}^{S}\pi_i^{1,1}F_i^{1,1}\left(t_{(k^{1,1})}^{1,1}\right) - \sum_{i=1}^{S}\pi_i^{1,0}F_i^{1,0}\left(t_{(k^{1,0})}^{1,0}\right) > \alpha\right) \\
&\leq \mathbb{P}\left(\sum_{i=1}^{S}\pi_i^{1,1}F_i^{1,1}\left(t_{i,\left(k_i^{1,1}+\lfloor\varepsilon n_i^{1,1}\rfloor\right)}^{1,1}\right) - \sum_{i=1}^{S}\pi_i^{1,0}F_i^{1,0}\left(t_{i,\left(k_i^{1,0}-\lfloor\varepsilon n_i^{1,0}\rfloor\right)}^{1,0}\right) > \alpha\right)
\end{aligned}$$

$\square$

## A.2 PROOF FOR PROPOSITION 3.3

*Proof for Proposition 3.3.* Note the classifier is

$$\phi = \begin{cases} \mathbb{1}\left\{f(x,0) > \hat{t}^{1,0}_{(k^{1,0})}\right\}, a = 0 \\ \mathbb{1}\left\{f(x,1) > \hat{t}^{1,1}_{(k^{1,1})}\right\}, a = 1 \end{cases}$$

So we can calculate the mis-classification error:

$$
\begin{aligned}
\mathbb{P}(Y \neq \hat{Y}) &= \mathbb{P}(Y = 1, \hat{Y} = 0) + \mathbb{P}(Y = 0, \hat{Y} = 1) \\
&= \mathbb{P}(Y = 1, \hat{Y} = 0, A = 0) + \mathbb{P}(Y = 1, \hat{Y} = 0, A = 1) \\
&\quad + \mathbb{P}(Y = 0, \hat{Y} = 1, A = 0) + \mathbb{P}(Y = 0, \hat{Y} = 1, A = 1) \\
&= \sum_{i=1}^{S} \pi_i \Big[ \mathbb{P}_i(Y = 1, \hat{Y} = 0, A = 0) + \mathbb{P}_i(Y = 1, \hat{Y} = 0, A = 1) + \\
&\quad \mathbb{P}_i(Y = 0, \hat{Y} = 1, A = 0) + \mathbb{P}_i(Y = 0, \hat{Y} = 1, A = 1) \Big]
\end{aligned}
\tag{13}
$$

For ecah specific i, we have

$$
\begin{aligned}
\mathbb{P}_i(Y = 1, \hat{Y} = 0, A = 0) &= \mathbb{P}_i(\hat{Y} = 1 \mid Y = 0, A = 0)\mathbb{P}_i(Y = 0, A = 0) \\
&= \mathbb{E}\left[\mathbb{P}_i\left(f(x,0) \leq \hat{t}^{1,0}_{(k^{1,0})} \mid Y = 1, A = 0\right) \mid \hat{t}^{1,0}_{(k^{1,0})}\right] p_0^i p_{Y,0}^i \\
&\leq \mathbb{E}\Big[\mathbb{P}_i\left(f(x,0) \leq t^{1,0}_{i,\left(\hat{k}_i^{1,0} + \lfloor \varepsilon n_i^{1,0}\rfloor + 1\right)} \mid Y = 1, A = 0\right) \\
&\quad \mid t^{1,0}_{i,\left(\hat{k}_i^{1,0} + \lfloor \varepsilon n_i^{1,0}\rfloor + 1\right)}\Big] p_0^i p_{Y,0}^i \\
&= \mathbb{E}\left[F_i^{1,0}\left(t^{1,0}_{i,\left(\hat{k}_i^{1,0} + \lfloor \varepsilon n_i^{1,0}\rfloor + 1\right)}\right) \mid t^{1,0}_{i,\left(\hat{k}_i^{1,0} + \lfloor \varepsilon n_i^{1,0}\rfloor + 1\right)}\right] p_0^i p_{Y,0}^i \\
&= \frac{\hat{k}_i^{1,0} + \lfloor \varepsilon n_i^{1,0}\rfloor + 1}{n_i^{1,0} + 1} p_0^i p_{Y,0}^i
\end{aligned}
$$

By the similar reasoning, we point out that

$$\mathbb{P}_i(Y = 1, \hat{Y} = 0, A = 0) \geq \frac{\hat{k}_i^{1,0} - \lfloor \varepsilon n_i^{1,0}\rfloor}{n_i^{1,0} + 1} p_0^i p_{Y,0}^i$$

and thus we have

$$\left| \mathbb{P}_i(Y = 1, \hat{Y} = 0, A = 0) - \frac{\hat{k}_i^{1,0} + 0.5}{n_i^{1,0} + 1} p_0^i p_{Y,0}^i \right| \leq \frac{\lfloor \varepsilon n_i^{1,0}\rfloor + 0.5}{n_i^{1,0} + 1} p_0^i p_{Y,0}^i \tag{14}$$

Moreover, we have

$$\mathbb{P}_i(Y=0, \hat{Y}=1, A=0) = \mathbb{P}_i(\hat{Y}=1 \mid Y=0, A=0)\mathbb{P}_i(Y=0, A=0)$$

$$= \mathbb{E}\left[\mathbb{P}_i\left(f(x,0) \geq \hat{t}_{(k^{1,0})}^{1,0} \mid Y=1, A=0\right) \mid \hat{t}_{(k^{1,0})}^{1,0}\right] p_0^i(1-p_{Y,0}^i)$$

$$\geq \mathbb{E}\Big[\mathbb{P}_i\Big(f(x,0) \geq t_{i,(\hat{k}_i^{0,0}+\lfloor \varepsilon n_i^{0,0}\rfloor+1)}^{0,0} \mid Y=1, A=0\Big)$$

$$\mid t_{i,(\hat{k}_i^{0,0}+\lfloor \varepsilon n_i^{0,0}\rfloor+1)}^{0,0}\Big] p_0^i(1-p_{Y,0}^i)$$

$$= \mathbb{E}\left[1 - F_i^{0,0}\left(t_{i,(\hat{k}_i^{0,0}+\lfloor \varepsilon n_i^{0,0}\rfloor+1)}^{0,0}\right) \mid t_{i,(\hat{k}_i^{0,0}+\lfloor \varepsilon n_i^{0,0}\rfloor+1)}^{0,0}\right] p_0^i(1-p_{Y,0}^i)$$

$$= \frac{n_i^{0,0} - \hat{k}_i^{0,0} - \lfloor \varepsilon n_i^{0,0}\rfloor}{n_i^{0,0}+1} p_0^i(1-p_{Y,0}^i)$$

Similar, we have

$$\mathbb{P}_i(Y=0, \hat{Y}=1, A=0) \leq \frac{n_i^{0,0} - \hat{k}_i^{0,0} + \lfloor \varepsilon n_i^{0,0}\rfloor + 1}{n_i^{0,0}+1} p_0^i(1-p_{Y,0}^i),$$

and combining these two result, we get

$$\left|\mathbb{P}_i(Y=0, \hat{Y}=1, A=0) - \frac{n_i^{0,0} - \hat{k}_i^{0,0} + 0.5}{n_i^{0,0}+1} p_0^i(1-p_{Y,0}^i)\right| \leq \frac{\lfloor \varepsilon n_i^{0,0}\rfloor + 0.5}{n_i^{0,0}+1} p_0^i(1-p_{Y,0}^i) \quad (15)$$

Following similar process of inequality 14 and 15, we can also show that

$$\left|\mathbb{P}_i(Y=1, \hat{Y}=0, A=1) - \frac{\hat{k}_i^{1,1} + 0.5\rfloor}{n_i^{1,1}+1} p_1^i p_{Y,1}^i\right| \leq \frac{\lfloor \varepsilon n_i^{1,1}\rfloor + 0.5}{n_i^{1,1}+1} p_1^i p_{Y,1}^i \quad (16)$$

$$\left|\mathbb{P}_i(Y=0, \hat{Y}=1, A=1) - \frac{n_i^{0,1} - \hat{k}_i^{0,1} + 0.5}{n_i^{0,1}+1} p_1^i(1-p_{Y,1}^i)\right| \leq \frac{\lfloor \varepsilon n_i^{0,1}\rfloor + 0.5}{n_i^{0,1}+1} p_1^i(1-p_{Y,1}^i) \quad (17)$$

Combining Inequality 14-17 into Equation 13, we complete our proof. $\qquad\square$

### A.3 PROOF FOR THEOREM 4.2

To begin with, the Fair Bayes-optimal Classifiers under Equality of Opportunity is defined by following lemma, wherein $\eta_a(x) := \mathbb{P}(Y=1 \mid A=a, X=x)$ stands for the proportion of group $Y=1$ conditioned on $A$ and $X$.

**Lemma A.2** (Theorem E.4 in (Zeng et al., 2022)). *Let $E^\star = \mathrm{DEOO}(f^\star)$. For any $\alpha > 0$, all fair Bayes-optimal classifiers $f_{E,\alpha}^\star$ under the fairness constraint $|\mathrm{DEOO}(f)| \leq \alpha$ are given as follows:*
*- When $|E^\star| \leq \alpha$, $f_{E,\alpha}^\star = f^\star$*

*- When $|E^\star| > \alpha$, suppose $\mathbb{P}_{X|A=1,Y=1}\left(\eta_1(X) = \frac{p_1 p_{Y,1}}{2(p_1 p_{Y,1} - t_{E,\alpha}^\star)}\right) = 0$, then for all $x \in \mathcal{X}$ and $a \in \mathcal{A}$,*

$$f_{E,\alpha}^\star(x,a) = I\left(\eta_a(x) > \frac{p_a p_{Y,a}}{2p_a p_{Y,a} + (1-2a)t_{E,\alpha}^\star}\right)$$

*where $t_{E,\alpha}^\star$ is defined as*

$$t_{E,\alpha}^\star = \sup\left\{t : \mathbb{P}_{Y|A=1,Y=1}\left(\eta_1(X) > \frac{p_1 p_{Y,1}}{2p_1 p_{Y,1} - t}\right)\right.$$

$$\left. > \mathbb{P}_{Y|A=0,Y=1}\left(\eta_0(X) > \frac{p_0 p_{Y,0}}{2p_0 p_{Y,0} + t}\right) + \frac{E^\star}{|E^\star|}\alpha\right\}.$$

**Lemma A.3** (Hoeffding's inequality). *Let $X_1, \ldots, X_n$ be independent random variables. Assume that $X_i \in [m_i, M_i]$ for every $i$. Then, for any $t > 0$, we have*

$$\mathbb{P}\left\{\sum_{i=1}^n (X_i - \mathbb{E}X_i) \geq t\right\} \leq e^{-\frac{2t^2}{\sum_{i=1}^n (M_i - m_i)^2}}$$

Then, we introduce several lemma to prove Theorem 4.2.

**Lemma A.4.** *For a distribution $F$ with a continuous density function, suppose $q(x)$ denotes the quantile of $x$ under $F$, then for $x > y$, we have $F_{(-)}(x - y) \leq q(x) - q(y) \leq F_{(+)}(x - y)$, where $F_{(-)}(x)$ and $F_{(+)}(x)$ are two monotonically increasing functions, $F_{(-)}(\epsilon) > 0, F_{(+)}(\epsilon) > 0$ for any $\epsilon > 0$ and $\lim_{\epsilon \to 0} F_{(-)}(\epsilon) = \lim_{\epsilon \to 0} F_{(+)}(\epsilon) = 0$.*

*Proof of Lemma A.4.* Since the domain of $q(x)$ is a closed set and $q(x)$ is continuous, we know that $q(x)$ is uniformly continuous. Thus we can easily find $F_{(+)}$ to satisfy the RHS. For $F_{(-)}$, we simply define $F_{(-)}(t) = \inf_x \{q(x + t) - q(t)\}$. Since $q(x + t) - q(t) > 0$ for $t > 0$ and the domain of $x$ is a closed set, we have $F_{(-)}(\epsilon) > 0$ for $\epsilon > 0$ and $\lim_{\epsilon \to 0} F_{(-)}(\epsilon) = 0$. Now we complete the proof. $\square$

*Proof for theorem 4.2.* In fact, (1) of the theorem is a direct application of Proposition 3.2, so we only need to prove (2). In partcular, the main idea of our proof is to find a bridge between fair Bayes optimal classifier and our output classifier.

To begin with, we show that there exist a classifier in our set which is quite similar with fair Bayes optimal classifier. Suppose the fair Bayes optimal classifier has the form $\phi_{\alpha'}^*(x, a) = \mathbb{I}\{f^*(x, a) > \lambda_a^*\}$ and our output classifier is of the form $\hat{\phi}(x, a) = \mathbb{1}\{f(x, a) > \lambda_a\}$.

For any $\epsilon > 0$, by Lemma A.4, we know that above than a positive probability $F_{i,(-)}^{1,a}(2\epsilon)$, $t_i^{1,a}$ would fall in the interval $[\lambda_a^* - \epsilon, \lambda_a^* + \epsilon]$ for each client $i$. Therefore, by the definition of $\varepsilon$-approximate quantile, we have at most with probability $\prod_{i=1}^S \left(1 - F_{i,(-)}^{1,0}(2\epsilon)\right)^{n_i^{1,0}} + \prod_{i=1}^S \left(1 - F_{i,(-)}^{1,1}(2\epsilon)\right)^{n_i^{1,1}}$, there exists $a \in \{0, 1\}$ such that all $t_{i,(k)}^{1,a}$ fall out of $[\lambda_a^* - \epsilon, \lambda_a^* + \epsilon]$. Thus, with probability $1 - \prod_{i=1}^S \left(1 - F_{i(-)}^{1,0}(2\epsilon)\right)^{n_i^{1,0}} - \prod_{i=1}^S \left(1 - F_{i(-)}^{1,1}(2\epsilon)\right)^{n_i^{1,1}}$, for $a \in \{0, 1\}$, there would exist i such that there exists at least one $t_i^{1,a}$ in $[\lambda_a^* - \epsilon, \lambda_a^* + \epsilon]$. So with $1 - \prod_{i=1}^S \left(1 - F_{i(-)}^{1,0}(2\epsilon)\right)^{n_i^{1,0}} - \prod_{i=1}^S \left(1 - F_{i(-)}^{1,1}(2\epsilon)\right)^{n_i^{1,1}} - \delta(n^{1,0}) - \delta(n^{1,1})$, there exist a classifier $\phi_0(x, a) = \mathbb{1}\left\{f(x, a) > \hat{t}^{1,a}\right\}$ such that $\hat{t}_*^{1,a} \in [\lambda_a^* - \epsilon - \gamma\varepsilon, \lambda_a^* + \epsilon + \gamma\varepsilon]$. We also denote $\phi_0^*(x, a) = \mathbb{1}\left\{f^*(x, a) > t_*^{1,a}\right\}$. Given the threshold is quite close, we further prove that the accuracy is quite close with a high probability. Actually, we have

$$
\begin{aligned}
&|\mathbb{P}\left(\phi_0(x, a) \neq Y\right) - \mathbb{P}\left(\phi_{\alpha'}^*(x, a) \neq Y\right)| \\
\leq& |\mathbb{P}\left(\phi_0(x, a) \neq Y\right) - \mathbb{P}\left(\phi_0^*(x, a) \neq Y\right)| + |\mathbb{P}\left(\phi_0^*(x, a) \neq Y\right) - \mathbb{P}\left(\phi_{\alpha'}^*(x, a) \neq Y\right)| \\
\leq& \mathbb{P}\left(t_*^{1,a} - \epsilon_0 \leq f^*(x, a) \leq t_*^{1,a} + \epsilon_0\right) + \mathbb{P}\left(\min\left\{t_*^{1,a}, \lambda_a^*\right\} \leq f^*(x, a) \leq \max\left\{t_*^{1,a}, \lambda_a^*\right\}\right) \quad (18) \\
\leq& F_{(+)}^*(2\epsilon_0) + F_{(+)}^*\left(\max\left\{t_*^{1,a}, \lambda_a^*\right\} - \min\left\{t_*^{1,a}, \lambda_a^*\right\}\right) \\
\leq& F_{(+)}^*(2\epsilon_0) + 2F_{(+)}^*(\epsilon + \gamma\varepsilon)
\end{aligned}
$$

with probability $1 - \prod_{i=1}^S \left(1 - F_{i,(-)}^{1,0}(2\epsilon)\right)^{n_i^{1,0}} - \prod_{i=1}^S \left(1 - F_{i,(-)}^{1,1}(2\epsilon)\right)^{n_i^{1,1}} - \delta(n^{1,0}) - \delta(n^{1,1})$.

Further we point out that

$$\| \text{DEOO}(\phi_0) | - | \text{DEOO}(\phi^*_{\alpha'}) \|$$
$$\leq \| \text{DEOO}(\phi_0) | - | \text{DEOO}(\phi^*_0) | + | DEOO(\phi^*_0) | - | \text{DEOO}(\phi^*_{\alpha'}) \|$$
$$= \| | \mathbb{P}(f > t^{1,0}_* \mid Y = 1, A = 0) - \mathbb{P}(f > t^{1,1}_* \mid Y = 1, A = 1) |$$
$$- | \mathbb{P}(f^* > t^{1,0}_* \mid Y = 1, A = 0) - \mathbb{P}(f^* > t^{1,1}_* \mid Y = 1, A = 1) | \|$$
$$+ \| | \mathbb{P}(f^* > t^{1,0}_* \mid Y = 1, A = 0) - \mathbb{P}(f^* > t^{1,1}_* \mid Y = 1, A = 1) |$$
$$- | \mathbb{P}(f^* > \lambda^*_0 \mid Y = 1, A = 0) - \mathbb{P}(f^* > \lambda^*_1 \mid Y = 1, A = 1) | \|$$
$$\leq | \mathbb{P}(f > t^{1,0}_* \mid Y = 1, A = 0) - \mathbb{P}(f^* > t^{1,0}_* \mid Y = 1, A = 0) |$$
$$+ | \mathbb{P}(f > t^{1,1}_* \mid Y = 1, A = 1) - \mathbb{P}(f^* > t^{1,1}_* \mid Y = 1, A = 1) |$$
$$+ \| | \mathbb{P}(f^* > t^{1,0}_* \mid Y = 1, A = 0) - \mathbb{P}(f^* > t^{1,1}_* \mid Y = 1, A = 1) |$$
$$- | \mathbb{P}(f^* > \lambda^*_0 \mid Y = 1, A = 0) - \mathbb{P}(f^* > \lambda^*_1 \mid Y = 1, A = 1) | \|$$
$$\leq \mathbb{P}(t^{1,0}_* - \epsilon_0 \leq f^*(x,a) \leq t^{1,0}_* + \epsilon_0) + \mathbb{P}(t^{1,1}_* - \epsilon_0 \leq f^*(x,a) \leq t^{1,1}_* + \epsilon_0)$$
$$+ | \mathbb{P}(f^* > t^{1,0}_* \mid Y = 1, A = 0) - \mathbb{P}(f^* > t^{1,1}_* \mid Y = 1, A = 1)$$
$$- \mathbb{P}(f^* > \lambda^*_0 \mid Y = 1, A = 0) + \mathbb{P}(f^* > \lambda^*_1 \mid Y = 1, A = 1) |$$
$$\leq 2 F^*_{(+)}(2\epsilon_0) + \mathbb{P}(\min\{t^{1,a}_*, \lambda^*_a\} \leq f^*(x,a) \leq \max\{t^{1,a}_*, \lambda^*_a\})$$
$$\leq 2 F^*_{(+)}(2\epsilon_0) + F^*_{(+)}(\max\{t^{1,a}_*, \lambda^*_a\} - \min\{t^{1,a}_*, \lambda^*_a\})$$
$$\leq 2 F^*_{(+)}(2\epsilon_0) + 2 F^*_{(+)}(\epsilon + \gamma\varepsilon)$$

Thus, we know that

$$|\text{DEOO}(\phi_0)| \leq | DEOO(\phi^*_{\alpha'}) | + 2 F^*_{(+)}(2\epsilon_0) + 2 F^*_{(+)}(\epsilon + \gamma\varepsilon)$$
$$= \alpha' + 2 F^*_{(+)}(2\epsilon_0) + 2 F^*_{(+)}(\epsilon + \gamma\varepsilon)$$

If $F^*_{(+)}(\epsilon + \gamma\varepsilon) \leq \frac{\alpha - \alpha'}{2} - F^*_{(+)}(2\epsilon_0)$, then there will exist at least one feasible classifier in the candidate set.

On the other hand, we could prove that the output classifier is quite similar with $\phi_0$ we mentioned above.

By Proposition 3.3, for any $\phi \in K$, we have

$$\left| \mathbb{P}(\phi(x,a) \neq Y) - \sum_{i=1}^{S} \pi_i \left[ \frac{\hat{k}_i^{1,0} + 0.5}{n_i^{1,0} + 1} p_0^i p_{Y,0}^i + \frac{\hat{k}_i^{1,1} + 0.5}{n_i^{1,1} + 1} p_1^i p_{Y,1}^i \right.\right. \tag{19}$$
$$\left.\left. + \frac{n_i^{0,0} + 0.5 - \hat{k}_i^{0,0}}{n_i^{0,0} + 1} p_0^i (1 - p_{Y,0}^i) + \frac{n_i^{0,1} + 0.5 - \hat{k}_i^{0,1}}{n_i^{0,1} + 1} p_1^i (1 - p_{Y,1}^i) \right] \right| \leq \theta$$

Therefore, we only need to check the influence induced by using $\hat{p}_a^i$ and $\hat{p}_{Y,a}^i$, instead of $p_0^i$ and $p_{Y,0}^i$. In detail, we point out this influence can be estimated by Hoeffding's inequality as follow:

Since $\hat{p}_a^i = \frac{n_i^{1,a} + n_i^{0,a}}{n_i}$ and $\hat{p}_{Y,a}^i = \frac{n_i^{1,a}}{n_i^{0,a} + n_i^{1,a}}$, we have $\frac{n_i^{1,a} + n_i^{0,a}}{n_i} = \frac{\sum_{j=1}^{n_i} \mathbb{1}\{Z_j^a = 1\}}{n}$ and $\frac{n_i^{1,a}}{n_i^{0,a} + n_i^{1,a}} = \frac{\sum_{j=1}^{n_i^{0,a} + n_i^{1,a}} \mathbb{1}\{Z_j^{Y,a} = 1\}}{n_i^{0,a} + n_i^{1,a}}$, where $Z_j^a \sim B(1, p_a^i)$ and $Z_j^{Y,a} \sim B(1, p_{Y,a}^i)$.

Thus, from Hoeffding's inequality, we have

$$\mathbb{P}\left( |\hat{p}_a^i - p_a^i| \geq \sqrt{\frac{n_i^{0,a}}{n_i}} \epsilon \right) \leq 2 e^{-2 n_i^{0,a} \epsilon^2}$$

For the same reason, we have we have

$$\mathbb{P}\left(\left|\hat{p}^i_{Y,a} - p^i_{Y,a}\right| \geq \sqrt{\frac{n^{0,a}_i}{n_i}}\epsilon\right) \leq 2e^{-2n^{0,a}_i\epsilon^2}$$

So, we have with probability $1 - 4\sum_{i=1}^S e^{-2n^{0,a}_i\epsilon^2}$

$$\begin{cases} \left|\hat{p}^i_a - p^i_a\right| \leq \sqrt{\dfrac{n^{0,a}_i}{n_i}}\epsilon \\[3mm] \left|\hat{p}^i_{Y,a} - p^i_{Y,a}\right| \leq \sqrt{\dfrac{n^{0,a}_i}{n^{*,a}_i}}\epsilon \end{cases},$$

where $n^{*,a}_i = (n^{0,a}_i + n^{1,a}_i)$.

Thus, with probability $1 - 4\sum_{i=1}^S (e^{-2n^{0,0}_i\epsilon^2} + e^{-2n^{0,1}_i\epsilon^2})$

$$\begin{aligned}
&\left|\mathbb{P}\left(\hat{\phi}_i(x,a) \neq Y\right) - \hat{\mathbb{P}}\left(\hat{\phi}_i(x,a) \neq Y\right)\right| \\
&\leq \left|\sum_{i=1}^S \pi^i\left[\frac{\hat{k}^{1,0}_i + 0.5}{n^{1,0}_i + 1}p^i_0 p^i_{Y,0} + \frac{\hat{k}^{1,1}_i + 0.5}{n^{1,1}_i + 1}p^i_1 p^i_{Y,1} + \frac{n^{0,0}_i + 0.5 - \hat{k}^{0,0}_i}{n^{0,0}_i + 1}p^i_0\left(1 - p^i_{Y,0}\right)\right.\right. \\
&\quad\left. + \frac{n^{0,1}_i + 0.5 - \hat{k}^{0,1}_i}{n^{0,1}_i + 1}p^i_1\left(1 - p^i_{Y,1}\right)\right] - \sum_{i=1}^S \pi^i\left[\frac{\hat{k}^{1,0}_i + 0.5}{n^{1,0}_i + 1}\hat{p}^i_0\hat{p}^i_{Y,0} + \frac{\hat{k}^{1,1}_i + 0.5}{n^{1,1}_i + 1}\hat{p}^i_1\hat{p}^i_{Y,1}\right. \\
&\quad\left.\left. + \frac{n^{0,0}_i + 0.5 - \hat{k}^{0,0}_i}{n^{0,0}_i + 1}\hat{p}^i_0\left(1 - \hat{p}^i_{Y,0}\right) + \frac{n^{0,1}_i + 0.5 - \hat{k}^{0,1}_i}{n^{0,1}_i + 1}\hat{p}^i_1\left(1 - \hat{p}^i_{Y,1}\right)\right]\right| \\
&\quad + \sum_{i=1}^S \pi_i\left[e^{0,0}_i p^i_0\left(1 - p^i_{Y,0}\right) + e^{0,1}_i p^i_0 p^i_{Y,0} + e^{1,0}_i p^i_1\left(1 - p^i_{Y,1}\right) + e^{1,1}_i p^i_1 p^i_{Y,1}\right] \\
&= |\sum_{i=1}^S \pi^i(A_i - \hat{A}_i)| + \sum_{i=1}^S \pi_i\left[e^{0,0}_i p^i_0\left(1 - p^i_{Y,0}\right) + e^{0,1}_i p^i_0 p^i_{Y,0} + e^{1,0}_i p^i_1\left(1 - p^i_{Y,1}\right) + e^{1,1}_i p^i_1 p^i_{Y,1}\right]
\end{aligned}$$
(20)

For $A_i - \hat{A}_i$, we have

$$
\begin{aligned}
A_i - \hat{A}_i \leq & \epsilon \left[ \sqrt{\frac{n_i^{0,0}}{n_i^{*,0}} \frac{\hat{k}_i^{1,0} + 0.5}{n^{1,0} + 1}} \left(p_0^i + p_{Y,0}^i\right) + \sqrt{\frac{n_i^{0,1}}{n_i^{*,1}} \frac{\hat{k}_i^{1,1} + 0.5}{n^{1,1} + 1}} \left(p_1^i + p_{Y,1}^i\right) \right] \\
& + \epsilon^2 \left( \frac{n_i^{0,0}}{n_i^{*,0}} \frac{\hat{k}_i^{1,0} + 0.5}{n^{1,0} + 1} + \frac{n_i^{0,1}}{n_i^{*,1}} \frac{\hat{k}_i^{1,1} + 0.5}{n^{1,1} + 1} \right) \\
& + \frac{n^{0,0} + 0.5 - \hat{k}_i^{0,0}}{n^{0,0} + 1} \sqrt{\frac{n_i^{0,0}}{n_i^{*,0}}} \epsilon \left[ \sqrt{\frac{n_i^{0,0}}{n_i^{*,0}}} \epsilon + p_0^i + p_{Y,0}^i + 1 \right] \\
& + \frac{n^{0,1} + 0.5 - \hat{k}_i^{0,1}}{n^{0,1} + 1} \sqrt{\frac{n_i^{0,1}}{n_i^{*,1}}} \epsilon \left[ \sqrt{\frac{n_i^{0,1}}{n_i^{*,1}}} \epsilon + p_1^i + p_{Y,1}^i + 1 \right] \\
\leq & \epsilon \left[ \sqrt{\frac{n_i^{0,0}}{n_i^{*,0}}} \left(p_0^i + p_{Y,0}^i\right) + \sqrt{\frac{n_i^{0,1}}{n_i^{*,1}}} \left(p_1^i + p_{Y,1}^i\right) \right] + \epsilon^2 \left( \frac{n_i^{0,0}}{n_i^{*,0}} + \frac{n_i^{0,1}}{n_i^{*,1}} \right) \\
& + \sqrt{\frac{n_i^{0,0}}{n_i^{*,0}}} \epsilon \left[ \sqrt{\frac{n_i^{0,0}}{n_i^{*,0}}} \epsilon + p_0^i + p_{Y,0}^i + 1 \right] \\
& + \sqrt{\frac{n_i^{0,1}}{n_i^{*,1}}} \epsilon \left[ \sqrt{\frac{n_i^{0,1}}{n_i^{*,1}}} \epsilon + p_1^i + p_{Y,1}^i + 1 \right] \\
\leq & 4\epsilon + 2\epsilon^2 + 2\epsilon^2 + 6\epsilon \\
= & 4\epsilon^2 + 10\epsilon
\end{aligned}
\tag{21}
$$

Combining Inequality 18-21, we complete the proof. $\qquad \square$

## A.4 DETAILED THEORY FOR LABEL SHIFT CASE

**Proposition A.1.** *Under Assumption 5.1, the misclassification error can be estimated by*

$$
\begin{aligned}
\hat{\mathbb{P}} \left( \hat{\phi}(x, a) \neq Y \right) = \sum_{i=1}^{S} \pi_i \Big[ & \frac{\hat{k}_i^{1,0} + 0.5}{n_i^{1,0} + 1} p_0^i p_{Y,0}^i w^{1,0} + \frac{\hat{k}_i^{1,1} + 0.5}{n_i^{1,1} + 1} p_1^i p_{Y,1}^i w^{1,1} \\
& + \frac{n_i^{0,0} + 0.5 - \hat{k}_i^{0,0}}{n_i^{0,0} + 1} p_0^i q_{Y,0}^i w^{0,0} + \frac{n_i^{0,1} + 0.5 - \hat{k}_i^{0,1}}{n_i^{0,1} + 1} p_1^i q_{Y,1}^i w^{0,1} \Big],
\end{aligned}
\tag{22}
$$

*where $w^{y,a} = \frac{p_a^{S+1} p_{Y,a}^{S+1}}{p_a p_{Y,a}}$. Further, discrepancy between empirical error and true error is limited by following inequality:*

$$
\left| \mathbb{P} \left( \hat{\phi}(x, a) \neq Y \right) - \hat{\mathbb{P}} \left( \hat{\phi}(x, a) \neq Y \right) \right| \leq \theta'
\tag{23}
$$

*where $e_i^{y,a} = \frac{2\lfloor \varepsilon n_i^{y,a} \rfloor + 1}{2 \left( n_i^{y,a} + 1 \right)}$ and $\theta' = \sum_{i=1}^{S} \pi_i \big[ e_i^{0,0} p_0^i q_{Y,0}^i w^{0,0} + e_i^{0,1} w^{0,1} p_0^i p_{Y,0}^i + e_i^{1,0} w^{1,0} p_1^i q_{Y,1}^i + e_i^{1,1} w^{1,1} p_1^i p_{Y,1}^i \big]$.*

*Proof for Proposition A.1.* Note the classifier is

$$
\phi = \begin{cases} \mathbb{1} \left\{ f(x,0) > \hat{t}_{(k^{1,0})}^{1,0} \right\}, a = 0 \\ \mathbb{1} \left\{ f(x,1) > \hat{t}_{(k^{1,1})}^{1,1} \right\}, a = 1 \end{cases}
$$

So we can calculate the mis-classification error in $P_{S+1}$. Denoted $\mathbb{P}_{S+1}$ the probability measure under the $P_{S+1}$ distribution, we have:

---

**Algorithm 2** FedFaiREE for label shift case

---

**Input:** Train dataset $D_i = D_i^{0,0} \cup D_i^{0,1} \cup D_i^{1,0} \cup D_i^{1,1}$; pre-trained classifier $\phi_0$ with function f; fainess constraint parameter $\alpha$ ; Confidence level parameter $\beta$; Weights of different clients $\pi$
**Output:** classifier $\hat{\phi}(x, a) = \mathbf{1}\{f(x, a) > t_{(k^{1,a})}^{1,a}\}$
**Client Side:**
**for** i=1,2,...,S **do**
  Score on train data points in $D_i$ and get $T_i^{y,a} = \{t_{i,1}^{y,a}, t_{i,2}^{y,a}, \cdots, t_{i,n_i^{y,a}}^{y,a}\}$
  Sort $T_i^{y,a}$
  Calculate q-digest of $T_i^{y,a}$ on client $i$
  Update digest to server
**end for**
**Server Side:**
Construct $K$ by $K = \{(\boldsymbol{k}^{1,0}, \boldsymbol{k}^{1,1})|L(\boldsymbol{k}^{1,0}, \boldsymbol{k}^{1,1}) < 1 - \beta\}$
Select optimal $(\boldsymbol{k}_0, \boldsymbol{k}_1)$ by minimizing equation 9 using estimated values $\hat{p}_a^i = \frac{n_i^{0,a} + n_i^{1,a}}{n_i^{0,0} + n_i^{0,1} + n_i^{1,0} + n_i^{1,1}}$ and $\hat{p}_{Y,a}^i = \frac{n_i^{1,a}}{n_i^{0,a} + n_i^{1,a}}$

---

$$
\begin{aligned}
&\mathbb{P}_{S+1}(Y \neq \hat{Y}) = \mathbb{P}_{S+1}(Y = 1, \hat{Y} = 0) + \mathbb{P}_{S+1}(Y = 0, \hat{Y} = 1) \\
&= \mathbb{P}_{S+1}(Y = 1, \hat{Y} = 0, A = 0) + \mathbb{P}_{S+1}(Y = 1, \hat{Y} = 0, A = 1) \\
&\quad + \mathbb{P}_{S+1}(Y = 0, \hat{Y} = 1, A = 0) + \mathbb{P}_{S+1}(Y = 0, \hat{Y} = 1, A = 1) \\
&= \mathbb{P}(Y = 1, \hat{Y} = 0, A = 0 \mid (X, Y, A) \sim P_{S+1}) + \mathbb{P}(Y = 1, \hat{Y} = 0, A = 1 \mid (X, Y, A) \sim P_{S+1}) \\
&\quad + \mathbb{P}(Y = 0, \hat{Y} = 1, A = 0 \mid (X, Y, A) \sim P_{S+1}) + \mathbb{P}(Y = 0, \hat{Y} = 1, A = 1 \mid (X, Y, A) \sim P_{S+1}) \\
&= \mathbb{P}(\hat{Y} = 0 \mid Y = 1, A = 0)p_0^{S+1}p_{Y,0}^{S+1} + \mathbb{P}(\hat{Y} = 0 \mid Y = 1, A = 1)p_1^{S+1}p_{Y,1}^{S+1} \\
&\quad + \mathbb{P}(\hat{Y} = 1 \mid Y = 0, A = 0)p_0^{S+1}(1 - p_{Y,0}^{S+1}) + \mathbb{P}(\hat{Y} = 1 \mid Y = 0, A = 1)p_1^{S+1}(1 - p_{Y,1}^{S+1}) \\
&= \sum_{i=1}^{S} \pi_i^{1,0}\mathbb{P}_i(\hat{Y} = 0 \mid Y = 1, A = 0)p_0^{S+1}p_{Y,0}^{S+1} + \sum_{i=1}^{S} \pi_i^{1,1}\mathbb{P}_i(\hat{Y} = 0 \mid Y = 1, A = 1)p_1^{S+1}p_{Y,1}^{S+1} \\
&\quad + \sum_{i=1}^{S} \pi_i^{0,0}\mathbb{P}(\hat{Y} = 1 \mid Y = 0, A = 0)p_0^{S+1}(1 - p_{Y,0}^{S+1}) \\
&\quad + \sum_{i=1}^{S} \pi_i^{0,1}\mathbb{P}(\hat{Y} = 1 \mid Y = 0, A = 1)p_1^{S+1}(1 - p_{Y,1}^{S+1}) \\
&= \sum_{i=1}^{S} \pi_i \Big[ w^{0,0}\mathbb{P}_i(Y = 1, \hat{Y} = 0, A = 0) + w^{0,1}\mathbb{P}_i(Y = 1, \hat{Y} = 0, A = 1) + \\
&\quad w^{1,0}\mathbb{P}_i(Y = 0, \hat{Y} = 1, A = 0) + w^{1,1}\mathbb{P}_i(Y = 0, \hat{Y} = 1, A = 1) \Big]
\end{aligned}
\tag{24}
$$

Then, since estimating $\mathbb{P}_i(Y = 0, \hat{Y} = y, A = a)$ shares similarities with the approach outlined in Proposition 3.3. This similarity in the estimation process allows us to successfully complete our proof. □

Given proof for Proposition A.1, proof for Theorem 5.2 is similar to Proof for Theorem 4.2

### A.5 DETAILED THEORY FOR DEO

**Proposition A.2.** *Under Assumption 3.1, for* $a \in \{0, 1\}$, *consider* $k^{1,a} \in \{1, \ldots, n^{1,a}\}$, *the corresponding* $\hat{k}_i^{1,a}$ *for* $i \in [S]$ *which are $\varepsilon$-approximate ranks and the score-based classifier*

$\phi(x,a) = \mathbb{1}\{f(x,a) > t^{1,a}_{(k^{1,a})}\}$ . *Define*

$$h_{y,a}(\boldsymbol{u}, \boldsymbol{v}) = \mathbb{P}\left(\sum_{i=1}^{S} \pi_i^{y,a} Q\left(u_i, n_i^{y,a} + 1 - u_i\right) - \sum_{i=1}^{S} \pi_i^{y,1-a} Q\left(v_i, n_i^{y,1-a} + 1 - v_i\right) \geq \alpha\right).$$

*Then we have:*

$$\mathbb{P}(|DEO(\phi)| \preceq (\alpha, \alpha)) \geq 1 - h^*_{1,1} - h^*_{1,0} - h^*_{0,1} - h^*_{0,0} \qquad (25)$$

*where the definitions of* $M_i^{y,a}$, $m_i^{y,a}$, $\pi_i^{y,a}$, $Q(A,B)$ *are similar to Proposition* 3.2, $h^*_{1,1} = h_{y,a}(\boldsymbol{M}^{y,a}, \boldsymbol{m}^{y,a})$

*Proof of Proposition* A.2. Note the output classifier is

$$\phi = \begin{cases} \mathbb{1}\left\{f(x,0) > \hat{t}^{1,0}_{(k^{1,0})}\right\}, a = 0 \\ \mathbb{1}\left\{f(x,1) > \hat{t}^{1,1}_{(k^{1,1})}\right\}, a = 1 \end{cases}$$

we have:

$$\begin{aligned}
\mathbb{P}(|DEO(\phi)| \preceq (\alpha, \alpha)) \geq{}& 1 - \mathbb{P}\left(\left|F^{1,1}\left(t^{1,1}_{(k^{1,1})}\right) - F^{1,0}\left(t^{1,0}_{(k^{1,0})}\right)\right| > \alpha\right) \\
& - \mathbb{P}\left(\left|F^{0,1}\left(t^{1,1}_{(k^{1,1})}\right) - F^{0,0}\left(t^{1,0}_{(k^{1,0})}\right)\right| > \alpha\right) \\
={}& 1 - \mathbb{P}\left(\sum_{i=1}^{S} \pi_i^{1,1} F_i^{1,1}\left(t^{1,1}_{(k^{1,1})}\right) - \sum_{i=1}^{S} \pi_i^{1,0} F_i^{1,0}\left(t^{1,0}_{(k^{1,0})}\right) > \alpha\right) \\
& - \mathbb{P}\left(\sum_{i=1}^{S} \pi_i^{1,1} F_i^{1,1}\left(t^{1,1}_{(k^{1,1})}\right) - \sum_{i=1}^{S} \pi_i^{1,0} F_i^{1,0}\left(t^{1,0}_{(k^{1,0})}\right) < -\alpha\right) \\
& - \mathbb{P}\left(\sum_{i=1}^{S} \pi_i^{0,1} F_i^{0,1}\left(t^{1,1}_{(k^{1,1})}\right) - \sum_{i=1}^{S} \pi_i^{0,0} F_i^{0,0}\left(t^{1,0}_{(k^{1,0})}\right) > \alpha\right) \\
& - \mathbb{P}\left(\sum_{i=1}^{S} \pi_i^{0,1} F_i^{0,1}\left(t^{1,1}_{(k^{1,1})}\right) - \sum_{i=1}^{S} \pi_i^{0,0} F_i^{0,0}\left(t^{1,0}_{(k^{1,0})}\right) < -\alpha\right)
\end{aligned}$$

The remainder of the proof is similar to the proof for Proposition 3.1

$\square$

Building upon Proposition A.2, we can further prove Theorem 5.4 using a similar approach as in Theorem 4.2.

# B  APPLICATION ON FURTHER NOTIONS

In this section, we delve into the application of FedFaiREE on additional fairness concepts.

## B.1  DEFINITION

To begin with, we introduce the definitions of various fairness concepts.

**Definition B.1** (Demographic Parity). *A classifier satisfies Demographic Parity if its prediction* $\widehat{Y}$ *is statistically independent of the sensitive attribute* $A$ *:*

$$\mathbb{P}(\widehat{Y} = 1 \mid A = 1) = \mathbb{P}(\widehat{Y} = 1 \mid A = 0)$$

**Definition B.2** (Predictive Equality). *A classifier satisfies Predictive Equality if it achieves the same TNR (or FPR) among protected groups:*

$$\mathbb{P}_{X|A=1,Y=0}(\widehat{Y} = 1) = \mathbb{P}_{X|A=0,Y=0}(\widehat{Y} = 1)$$

---

**Algorithm 3** FedFaiREE for DEO

---

**Input:** Train dataset $D_i = D_i^{0,0} \cup D_i^{0,1} \cup D_i^{1,0} \cup D_i^{1,1}$; pre-trained classifier $\phi_0$ with function f; fairness constraint parameter $\alpha$ ; Confidence level parameter $\beta$; Weights of different clients $\pi$
**Output:** classifier $\hat{\phi}(x,a) = \mathbf{1}\{f(x,a) > t^{1,a}_{(k^{1,a})}\}$
**Client Side:**
**for** i=1,2,...,S **do**
    Score on train data points in $D_i$ and get $T_i^{y,a} = \{t_{i,1}^{y,a}, t_{i,2}^{y,a}, \cdots, t_{i,n_i^{y,a}}^{y,a}\}$
    Sort $T_i^{y,a}$
    Calculate q-digest of $T_i^{y,a}$ on client $i$
    Update digest to server
**end for**
**Server Side:**
Construct $K$ by $K = \{(\boldsymbol{k}^{1,0}, \boldsymbol{k}^{1,1}) | L(\boldsymbol{k}^{1,0}, \boldsymbol{k}^{1,1}) < 1 - \beta\}$, where L is defined in Equation 11
Select optimal $(\boldsymbol{k}_0, \boldsymbol{k}_1)$ by minimizing equation 5 using estimated values $\hat{p}_a^i = \frac{n_i^{0,a} + n_i^{1,a}}{n_i^{0,0} + n_i^{0,1} + n_i^{1,0} + n_i^{1,1}}$ and $\hat{p}_{Y,a}^i = \frac{n_i^{1,a}}{n_i^{0,a} + n_i^{1,a}}$

---

**Definition B.3** (Equalized Accuracy). *A classifier satisfies Equalized Accuracy if its misclassification error is statistically independent of the sensitive attribute A:*

$$\mathbb{P}(\widehat{Y} \neq Y \mid A = 1) = \mathbb{P}(\widehat{Y} \neq Y \mid A = 0)$$

Similar to $DEOO$ and $DEO$, we define the following indicators:

$$\text{DDP} = \mathbb{P}_{X|A=1}(\widehat{Y} = 1) - \mathbb{P}_{X|A=0}(\widehat{Y} = 1) \tag{26}$$

$$\text{DPE} = \mathbb{P}_{X|A=1,Y=0}(\widehat{Y} = 1) - \mathbb{P}_{X|A=0,Y=0}(\widehat{Y} = 1) \tag{27}$$

$$\text{DEA} = \mathbb{P}(\widehat{Y} \neq Y \mid A = 1) - \mathbb{P}(\widehat{Y} \neq Y \mid A = 0). \tag{28}$$

## B.2 THEORY AND ALGORITHM

Similar to $DEO$ and $DEOO$, we To be concise, we denote $n_i^{*,a}$ as denotes the size of subset of dataset $D_i$ that satisfies $A = a$. Similar explanations apply to $k^{*,a}$.

### B.2.1 FEDFAIREE FOR DDP

**Proposition B.1.** *Under Assumption 3.1, for $a \in \{0,1\}$, consider $k^{*,a} \in \{1,\ldots,n^{*,a}\}$, the corresponding $\hat{k}_i^{*,a}$ for $i \in [S]$ which are $\varepsilon$-approximate ranks and the score-based classifier $\phi(x,a) = \mathbb{1}\{f(x,a) > t^{*,a}_{(k^{*,a})}\}$ . Define*

$$h_{*,a}(\boldsymbol{u}, \boldsymbol{v}) = \mathbb{P}\left(\sum_{i=1}^{S} \pi_i^{*,a} Q\left(u_i, n_i^{*,a} + 1 - u_i\right) - \sum_{i=1}^{S} \pi_i^{*,1-a} Q\left(v_i, n_i^{*,1-a} + 1 - v_i\right) \geq \alpha\right).$$

*Then we have:*

$$\mathbb{P}(|DDP(\phi)| > \alpha) \leq h_{*,0}(\boldsymbol{M}^{*,0}, \boldsymbol{m}^{*,1}) + h_{*,1}(\boldsymbol{M}^{*,1}, \boldsymbol{m}^{*,0}) \tag{29}$$

*Where $\pi_i^{*,a} = \mathbb{P}(sampling\ x\ from\ client\ i \mid sampling\ x\ with\ sensitive\ attribute A = a)$, $M_i^{*,a} = max\left(\lceil \hat{k}_i^{*,a} + \varepsilon n_i^{*,a}\rceil, n_i^{*,a} + 1\right)$, $m_i^{*,a} = min\left(\lceil \hat{k}_i^{*,a} - \varepsilon n_i^{*,a}\rceil, 0\right)$, and $Q(A,B)$ are independent random variables following Beta distribution, $Q(A,B) \sim Beta(A,B)$. Especially, we define $Q(0,B) = 0$ and $Q(A,0) = 1$ for $A, B \neq 0$.*

**Theorem B.4.** *Under Assumption 3.1 and 4.1, given $\alpha' < \alpha$. Suppose $\hat{\phi}$ is the final output of FedFaiREE, we have:*

*(1) $|DDP(\hat{\phi})| < \alpha$ with probability $(1 - \delta)^N$, where $N$ is the size of the candidate set.*

---

**Algorithm 4** FedFaiREE for DDP

---

**Input:** Train dataset $D_i = D_i^{0,0} \cup D_i^{0,1} \cup D_i^{1,0} \cup D_i^{1,1}$; pre-trained classifier $\phi_0$ with function f; fairness constraint parameter $\alpha$ ; Confidence level parameter $\beta$; Weights of different clients $\pi$
**Output:** classifier $\hat{\phi}(x, a) = \mathbf{1}\{f(x,a) > t_{(k^{1,a})}^{1,a}\}$
**Client Side:**
**for** i=1,2,...,S **do**
    Score on train data points in $D_i$ and get $T_i^{y,a} = \{t_{i,1}^{y,a}, t_{i,2}^{y,a}, \cdots, t_{i,n_i^{y,a}}^{y,a}\}$
    Sort $T_i^{y,a}$
    Calculate q-digest of $T_i^{y,a}$ on client $i$
    Update digest to server
**end for**
**Server Side:**
Construct $K$ by $K = \{(\boldsymbol{k}^{1,0}, \boldsymbol{k}^{1,1}) | L(\boldsymbol{k}^{1,0}, \boldsymbol{k}^{1,1}) < 1 - \beta\}$, where L is defined by the right-hand side of Inequality 29
Select optimal $(\boldsymbol{k}_0, \boldsymbol{k}_1)$ by minimizing equation 5 using estimated values $\hat{p}_a^i = \frac{n_i^{0,a} + n_i^{1,a}}{n_i^{0,0} + n_i^{0,1} + n_i^{1,0} + n_i^{1,1}}$ and $\hat{p}_{Y,a}^i = \frac{n_i^{1,a}}{n_i^{0,a} + n_i^{1,a}}$

---

*(2) Suppose the density distribution functions of $f^*$ under $A = a, Y = 1$ are continuous. When the input classifier $f$ satisfies $|f(x,a) - f^*(x,a)| \le \epsilon_0$, for any $\epsilon > 0$ such that $F_{(+)}^*(\epsilon + \gamma\varepsilon) \le \frac{\alpha - \alpha'}{2} - F_{(+)}^*(2\epsilon_0)$, we have*

$$\mathbb{P}(\hat{\phi}(x,a) \ne Y) - \mathbb{P}(\phi_{\alpha'}^*(x,a) \ne Y) \le 2F_{(+)}^*(2\epsilon_0) + 2F_{(+)}^*(\epsilon + \gamma\varepsilon) + 8\epsilon^2 + 20\epsilon + 2\theta \quad (30)$$

*with probability $1 - 4\sum_{i=1}^S (e^{-2n_i^{0,0}\epsilon^2} + e^{-2n_i^{0,1}\epsilon^2}) - \prod_{i=1}^S \left(1 - F_{i(-)}^{1,0}(2\epsilon)\right)^{n_i^{1,0}} - \prod_{i=1}^S \left(1 - F_{i(-)}^{1,1}(2\epsilon)\right)^{n_i^{1,1}} - \delta$, where $\delta = \delta^{1,0}(n^{1,0}) + \delta^{1,1}(n^{1,1})$, $\theta$ is defined in Proposition3.3 and the definition of $F_{(+)}$ and $F_{(-)}$ are shown in Lemma A.4*

### B.2.2 FEDFAIREE FOR DPE

**Proposition B.2.** *Under Assumption 3.1, for $a \in \{0, 1\}$, consider $k^{0,a} \in \{1, \ldots, n^{0,a}\}$, the corresponding $\hat{k}_i^{0,a}$ for $i \in [S]$ which are $\varepsilon$-approximate ranks and the score-based classifier $\phi(x, a) = \mathbb{1}\{f(x,a) > t_{(k^{0,a})}^{0,a}\}$. Define*

$$h_{y,a}(\boldsymbol{u}, \boldsymbol{v}) = \mathbb{P}\left(\sum_{i=1}^S \pi_i^{y,a} Q\left(u_i, n_i^{y,a} + 1 - u_i\right) - \sum_{i=1}^S \pi_i^{y,1-a} Q\left(v_i, n_i^{y,1-a} + 1 - v_i\right) \ge \alpha\right).$$

*Then we have:*

$$\mathbb{P}(|DPE(\phi)| > \alpha) \le h_{0,1}(\boldsymbol{M}^{0,1}, \boldsymbol{m}^{0,0}) + h_{0,0}(\boldsymbol{M}^{0,0}, \boldsymbol{m}^{0,0}) \quad (31)$$

*where $M_i^{0,a} = \lceil \hat{k}_i^{0,a} + \varepsilon n_i^{0,a} \rceil$, $m_i^{0,a} = \lceil \hat{k}_i^{0,a} - \varepsilon n_i^{0,a} \rceil$, $\pi_i^{y,a} = \mathbb{P}($sampling $x$ from client $i$ | sampling $x$ with label $Y = y$ and $A = a$), and $Q(A, B)$ are independent random variables following Beta distribution, $Q(A, B) \sim Beta(A, B)$.*

**Theorem B.5.** *Under Assumption 3.1 and 4.1, given $\alpha' < \alpha$. Suppose $\hat{\phi}$ is the final output of FedFaiREE, we have:*

*(1) $|DPE(\hat{\phi})| < \alpha$ with probability $(1 - \delta)^N$, where $N$ is the size of the candidate set.*

*(2) Suppose the density distribution functions of $f^*$ under $A = a, Y = 1$ are continuous. When the input classifier $f$ satisfies $|f(x,a) - f^*(x,a)| \le \epsilon_0$, for any $\epsilon > 0$ such that $F_{(+)}^*(\epsilon + \gamma\varepsilon) \le \frac{\alpha - \alpha'}{2} - F_{(+)}^*(2\epsilon_0)$, we have*

$$\mathbb{P}(\hat{\phi}(x,a) \ne Y) - \mathbb{P}(\phi_{\alpha'}^*(x,a) \ne Y) \le 2F_{(+)}^*(2\epsilon_0) + 2F_{(+)}^*(\epsilon + \gamma\varepsilon) + 8\epsilon^2 + 20\epsilon + 2\theta \quad (32)$$

---

**Algorithm 5** FedFaiREE for DPE

---

**Input:** Train dataset $D_i = D_i^{0,0} \cup D_i^{0,1} \cup D_i^{1,0} \cup D_i^{1,1}$; pre-trained classifier $\phi_0$ with function f; fairness constraint parameter $\alpha$ ; Confidence level parameter $\beta$; Weights of different clients $\pi$
**Output:** classifier $\hat{\phi}(x,a) = \mathbf{1}\{f(x,a) > t_{(k^{1,a})}^{1,a}\}$
**Client Side:**
**for** i=1,2,...,S **do**
    Score on train data points in $D_i$ and get $T_i^{y,a} = \{t_{i,1}^{y,a}, t_{i,2}^{y,a}, \cdots, t_{i,n_i^{y,a}}^{y,a}\}$
    Sort $T_i^{y,a}$
    Calculate q-digest of $T_i^{y,a}$ on client $i$
    Update digest to server
**end for**
**Server Side:**
Construct $K$ by $K = \{(\boldsymbol{k}^{1,0}, \boldsymbol{k}^{1,1}) | L(\boldsymbol{k}^{1,0}, \boldsymbol{k}^{1,1}) < 1 - \beta\}$, where L is defined by the right-hand side of Inequality 31
Select optimal $(\boldsymbol{k}_0, \boldsymbol{k}_1)$ by minimizing equation 5 using estimated values $\hat{p}_a^i = \frac{n_i^{0,a} + n_i^{1,a}}{n_i^{0,0} + n_i^{0,1} + n_i^{1,0} + n_i^{1,1}}$ and $\hat{p}_{Y,a}^i = \frac{n_i^{1,a}}{n_i^{0,a} + n_i^{1,a}}$

---

with probability $1 - 4\sum_{i=1}^{S}(e^{-2n_i^{0,0}\epsilon^2} + e^{-2n_i^{0,1}\epsilon^2}) - \prod_{i=1}^{S}\left(1 - F_{i(-)}^{1,0}(2\epsilon)\right)^{n_i^{1,0}} - \prod_{i=1}^{S}\left(1 - F_{i(-)}^{1,1}(2\epsilon)\right)^{n_i^{1,1}} - \delta$, where $\delta = \delta^{1,0}(n^{1,0}) + \delta^{1,1}(n^{1,1})$, $\theta$ is defined in Proposition 3.3 and the definition of $F_{(+)}$ and $F_{(-)}$ are shown in Lemma A.4

### B.2.3 FedFaiREE for DEA

**Proposition B.3.** *Under Assumption 3.1, for $a \in \{0,1\}$, consider $k^{y,a} \in \{1, \ldots, n^{y,a}\}$, the corresponding $\hat{k}_i^{y,a}$ for $i \in [S]$ which are $\varepsilon$-approximate ranks and the score-based classifier $\phi(x,a) = \mathbb{1}\{f(x,a) > t_{(k^{1,a})}^{1,a}\}$. Define*

$$
\begin{aligned}
h_{*,a}(\boldsymbol{u}^1, \boldsymbol{u}^0, \boldsymbol{v}^1, \boldsymbol{v}^0) = \mathbb{P}\bigg( & p_{y,a} - p_{y,1-a} - p_{y,a}\sum_{i=1}^{S}\pi_i^{1,a}Q\left(u_i^1, n_i^{1,a} + 1 - u_i^1\right) \\
& + (1 - p_{y,a})\sum_{i=1}^{S}\pi_i^{0,a}Q\left(u_i^0, n_i^{0,a} + 1 - u_i^0\right) \\
& + p_{y,1-a}\sum_{i=1}^{S}\pi_i^{1,1-a}Q\left(v_i^1, n_i^{1,1-a} + 1 - v_i^1\right) \\
& - (1 - p_{y,1-a})\sum_{i=1}^{S}\pi_i^{0,1-a}Q\left(v_i^0, n_i^{0,1-a} + 1 - v_i^0\right) \geq \alpha\bigg).
\end{aligned}
$$

*Then we have:*

$$
\mathbb{P}(|DPE(\phi)| > \alpha) \leq h_{*,1}(\boldsymbol{m}^{1,1}, \boldsymbol{M}^{0,1}, \boldsymbol{M}^{1,0}, \boldsymbol{m}^{0,0}) + h_{*,0}(\boldsymbol{m}^{1,0}, \boldsymbol{M}^{0,0}, \boldsymbol{M}^{1,1}, \boldsymbol{m}^{0,1}) \quad (33)
$$

*where $M_i^{0,a} = \lceil \hat{k}_i^{0,a} + \varepsilon n_i^{0,a} \rceil$, $m_i^{0,a} = \lceil \hat{k}_i^{0,a} - \varepsilon n_i^{0,a} \rceil$, $\pi_i^{y,a} = \mathbb{P}(sampling\ x\ from\ client\ i\ |\ sampling\ x\ with\ label\ Y = y\ and\ A = a)$, and $Q(A,B)$ are independent random variables following Beta distribution, $Q(A,B) \sim Beta(A,B)$.*

**Theorem B.6.** *Under Assumption 3.1 and 4.1, given $\alpha' < \alpha$. Suppose $\hat{\phi}$ is the final output of FedFaiREE, we have:*

*(1) $|DEA(\hat{\phi})| < \alpha$ with probability $(1 - \delta)^N$, where $N$ is the size of the candidate set.*

*(2) Suppose the density distribution functions of $f^*$ under $A = a, Y = 1$ are continuous. When the input classifier $f$ satisfies $|f(x,a) - f^*(x,a)| \leq \epsilon_0$, for any $\epsilon > 0$ such that $F_{(+)}^*(\epsilon + \gamma\varepsilon) \leq$*

---

**Algorithm 6** FedFaiREE for DEA

---

**Input:** Train dataset $D_i = D_i^{0,0} \cup D_i^{0,1} \cup D_i^{1,0} \cup D_i^{1,1}$; pre-trained classifier $\phi_0$ with function f; fairness constraint parameter $\alpha$ ; Confidence level parameter $\beta$; Weights of different clients $\pi$

**Output:** classifier $\hat{\phi}(x,a) = \mathbf{1}\{f(x,a) > t_{(k^{1,a})}^{1,a}\}$

**Client Side:**

**for** i=1,2,...,S **do**

    Score on train data points in $D_i$ and get $T_i^{y,a} = \{t_{i,1}^{y,a}, t_{i,2}^{y,a}, \cdots, t_{i,n_i^{y,a}}^{y,a}\}$

    Sort $T_i^{y,a}$

    Calculate q-digest of $T_i^{y,a}$ on client $i$

    Update digest to server

**end for**

**Server Side:**

Construct $K$ by $K = \{(\boldsymbol{k}^{1,0}, \boldsymbol{k}^{1,1}) | L(\boldsymbol{k}^{1,0}, \boldsymbol{k}^{1,1}) < 1 - \beta\}$, where L is defined by the right-hand side of Inequality 33

Select optimal $(\boldsymbol{k}_0, \boldsymbol{k}_1)$ by minimizing equation 5 using estimated values $\hat{p}_a^i = \frac{n_i^{0,a} + n_i^{1,a}}{n_i^{0,0} + n_i^{0,1} + n_i^{1,0} + n_i^{1,1}}$ and $\hat{p}_{Y,a}^i = \frac{n_i^{1,a}}{n_i^{0,a} + n_i^{1,a}}$

---

$\frac{\alpha - \alpha'}{2} - F_{(+)}^*(2\epsilon_0)$, we have

$$\mathbb{P}(\hat{\phi}(x,a) \neq Y) - \mathbb{P}(\phi_{\alpha'}^*(x,a) \neq Y) \leq 2F_{(+)}^*(2\epsilon_0) + 2F_{(+)}^*(\epsilon + \gamma\varepsilon) + 8\epsilon^2 + 20\epsilon + 2\theta \quad (34)$$

with probability $1 - 4\sum_{i=1}^S (e^{-2n_i^{0,0}\epsilon^2} + e^{-2n_i^{0,1}\epsilon^2}) - \prod_{i=1}^S \left(1 - F_{i(-)}^{1,0}(2\epsilon)\right)^{n_i^{1,0}} - \prod_{i=1}^S \left(1 - F_{i(-)}^{1,1}(2\epsilon)\right)^{n_i^{1,1}} - \delta$, where $\delta = \delta^{1,0}(n^{1,0}) + \delta^{1,1}(n^{1,1})$, $\theta$ is defined in Proposition 3.3 and the definition of $F_{(+)}$ and $F_{(-)}$ are shown in Lemma A.4

### B.3 CONNECTION WITH FAIRNESS METRICS IN (HU ET AL., 2022) AND (PAPADAKI ET AL., 2022)

Hu et al. (2022) introduces several group fairness metrics as follow:

**Definition B.7.** *A classifier h satisfies Bounded Group Loss (BGL) at level $\zeta$ under distribution $\mathcal{D}$ if for all $a \in A$, we have $\mathbb{E}[l(h(x), y) \mid A = a] \leq \zeta$.*

**Definition B.8.** *A classifier h satisfies Conditional Bounded Group Loss (CBGL) for $y \in Y$ at level $\zeta_y$ under distribution $\mathcal{D}$ if for all $a \in A$, we have $\mathbb{E}[l(h(x), y) \mid A = a, Y = y] \leq \zeta_y$.*

When considering y as a binary variable and the loss function l being the 0-1 loss function, BGL is equivalent to

$$\mathbb{P}[\hat{y} \neq y| \mid A = a] \leq \zeta,$$

holding for any a, whereas Demographic Parity refers to

$$\mathbb{P}[\hat{y} \neq y| \mid A = 0] = \mathbb{P}[\hat{y} \neq y| \mid A = 1].$$

In this context, BGL can be understood as a relaxation of Demographic Parity.

Similarly, when considering y as a binary variable and the loss function l being the 0-1 loss function, CBGL is equivalent to

$$\mathbb{P}[\hat{y} \neq y| \mid A = a, Y = y] \leq \zeta_y,$$

holding for any a, whereas Equalized Odds refers to

$$\mathbb{P}[\hat{y} \neq y| \mid A = 0, Y = y] = \mathbb{P}[\hat{y} \neq y| \mid A = 1, Y = y].$$

In this context, CBGL can be understood as a relaxation of Equalized Odds.

According to (Hu et al., 2022), the metric that Papadaki et al. (2022) considers is equivalent to

**Definition B.9.** *FedMinMax(Papadaki et al., 2022) aims to solve for the following objective:* $\min_h \max_{\boldsymbol{\lambda} \in \mathbb{R}_+^{|A|}, \|\boldsymbol{\lambda}\|_1 = 1} \sum_{a \in A} \boldsymbol{\lambda}_a \mathbf{r}_a(h)$, *where* $\mathbf{r}_a(h) \quad := \quad \sum_{k=1}^{K} \mathbf{r}_{a,k}(h) \quad = \sum_{k=1}^{K} \left( 1/m_a \sum_{a_{k,i}=a} l\left(h\left(x_{k,i}\right), y_{k,i}\right)\right)$, *K stands for client number and $m_a$ stands for numbers of points with attribute a.*

Similarly, this can be understood as a relaxation of Demographic Parity in the context of considering y as a binary variable and the loss function l being the 0-1 loss function.

## C EXPERIMENT DETAILS

### C.1 FURTHER SELECTION IN CANDIDATE SET CONSTRUCTION

To further simplify the candidate set selection, similar to FaiREE(Li et al., 2022), we note that, by Lemma A.2, if we assume our input classifier $f$ is similar to $f^*$, we have

$$t_a = \frac{p_a p_{Y,a}}{2p_a p_{Y,a} + (1 - 2a)t_{E,\alpha}^\star}, \tag{35}$$

which means

$$t_{E,\alpha}^\star = \frac{p_a p_{Y,a} - 2p_a p_{Y,a} t_a}{(1 - 2a)t_a} \tag{36}$$

Therefore, bringing Equation 36 ($a = 0$) into Equation 35 ($a = 1$), we have

$$t_0 = \frac{p_0 p_{Y,0}}{2p_0 p_{Y,0} + 2p_1 p_{Y,1} - p_1 p_{Y,1}/t_1} \tag{37}$$

This inspired us that we could further simplify the construction of candidate set K by replacing Equation 4 with

$$K = \{(\boldsymbol{k}^{1,0}, \boldsymbol{k}^{1,1}) | L(\boldsymbol{k}^{1,0}, \boldsymbol{k}^{1,1}) < 1 - \beta, k^{1,0} = \mu(k^{1,1})\}, \tag{38}$$

Where $\mu(k_1) = \arg\min_{k_0} \frac{p_0 p_{Y,0}}{2p_0 p_{Y,0} + 2p_1 p_{Y,1} - p_1 p_{Y,1}/\hat{t}_{k_1}}$

### C.2 MODEL DETAILS AND HYPERPARAMETER SELECTION

We employed several existing Federated Learning models in the experiment, and their detailed information is listed as follows:

1. FedAvg(McMahan et al., 2017): FedAvg is a fundamental Federated Learning model that serves as the foundational baseline for our experiments. It operates by computing model updates on each client's local data and then aggregates these updates on a central server through averaging. FedAvg doesn't specifically address fairness concerns but is crucial for benchmarking purposes.

2. AFL(Mohri et al., 2019): AFL, short for Agnostic federated learning, is a framework that focuses on improving fairness and robustness within the Federated Learning paradigm. AFL achieves this by optimizing the centralized model for any target distribution formed by a mixture of client distributions. It aims to avoid favoring any specific client and instead performs well on any possible combination of them.

3. FedFB(Zeng et al., 2021): FedFB is a novel framework designed for fairness-aware Federated Learning. Drawing inspiration from FairBatch, a fairness algorithm for centralized data, FedFB extends this concept to the Federated Learning setting. It incorporates both local debiasing and global reweighting for each client within the framework to achieve fairness objectives.

4. FairFed(Ezzeldin et al., 2023): FairFed is another innovative framework for fairness-aware Federated Learning. It employs a unique approach to improving fairness by reweighting clients based on updated local fairness indicators during each epoch. This allows FairFed to combine multiple local debiasing methods effectively.

To compare performance in terms of DEOO, we selected FedFB with respect to Equal Opportunity (EO) as presented in Zeng et al. (2021), and FairFed-FB-EO from FairFed as introduced in Ezzeldin et al. (2023). These are specific models within the FedFB and FairFed frameworks that are designed for DEOO.

We also note that there are concerns raised by the fairness community regarding the COMPAS dataset underscore crucial complexities within algorithmic fairness research(Bao et al., 2021). While Risk Assessment Instrument (RAI) datasets like COMPAS serve as prevalent benchmarks, their oversimplification of the intricate dynamics within real-world criminal justice processes poses significant challenges. Measurement biases and errors inherent in pretrial RAI datasets limit the direct translation of fairness claims to actual outcomes within the criminal justice system. Additionally, the technical focus on these data as a benchmark sometimes ignores the contextual grounding necessary for working with RAI datasets. Ethical reflection within socio-technical systems further highlights the necessity of acknowledging and grappling with the limitations and complexities inherent in RAI datasets.

Additionally, the hyperparameter selection ranges for each model are shown in Table 2.

**Table 2:** Hyperparameter Selection Ranges

| Model | Hyperparameter | Ranges |
|---|---|---|
| General | Learning rate | {0.001, 0.005, 0.01} |
| | Global round | {5, 10, 20, 30, 40, 50, 80} |
| | Local round | {5, 10} |
| | Local batch size | {16, 32, 64, 128} |
| | Hidden layer | {5, 10, 50} |
| | Optimizer | {Adam, Sgd} |
| | Fraction | {1} |
| | Parameter for Dirichlet distribution | {1} for Adult, {10} for Compas |
| | Number of Clients | {100} for Adult, {10} for Compas |
| | Sensitive Group | Female |
| FedFaiREE | Confidence level | {95%} |
| Qdigest | Accuracy | {$1/2^7$} for Adult, {$1/2^{10}$} for Compas |
| | Compression factor | {300} for Adult, {150} for Compas |
| AFL | Step size ($\gamma$) | {0.005, 0.01, 0.05} |
| FedFB | Step size ($\alpha$) | {0.005, 0.01, 0.05} |
| FairFed | Global step size ($\beta$) | {0.005, 0.01, 0.05} |
| | Local debiasing step size ($\alpha$) | {0.005, 0.01, 0.05} |

We further present a data split sample in Table 3, where random seed was set to be 0.

## C.3 MORE DETAILED RESULTS

In this subsection, we present a more detailed analysis of the experimental results from Section 6. Table 4 and Table 5 respectively illustrate the variances in the results obtained from the Adult dataset and the Compas dataset.

Table 3: **Heterogeneous data distribution on the sensitive attribute.** The client index is sorted by number of Male.

| Minimum ten clients | | | Maximum ten clients | | |
|---|---|---|---|---|---|
| Client id | Male | Female | Client id | Male | Female |
| 1 | 6 | 41 | 91 | 738 | 118 |
| 2 | 6 | 117 | 92 | 863 | 49 |
| 3 | 6 | 297 | 93 | 880 | 52 |
| 4 | 13 | 35 | 94 | 956 | 147 |
| 5 | 20 | 310 | 95 | 961 | 50 |
| 6 | 22 | 120 | 96 | 1101 | 35 |
| 7 | 24 | 234 | 97 | 1245 | 102 |
| 8 | 30 | 70 | 98 | 1250 | 31 |
| 9 | 32 | 124 | 99 | 1277 | 180 |
| 10 | 33 | 26 | 100 | 1480 | 24 |

Table 4: **Results with standard deviation on Adult.**

| Model | FedFaiREE | $\alpha$ | $\overline{ACC}$ | $\overline{|DEOO|}$ | $|DEOO|_{95}$ |
|---|---|---|---|---|---|
| | | | **Adult** | | |
| **FedAvg** | ✗ | / | 0.844 (0.003) | 0.131 (0.030) | 0.178 |
| | ✓ | 0.10 | 0.843 (0.003) | **0.038** (0.026) | **0.083** |
| **AFL** | ✗ | / | 0.848 (0.004) | 0.101 (0.040) | 0.169 |
| | ✓ | 0.10 | 0.848 (0.004) | **0.034** (0.027) | **0.081** |
| **FedFB** | ✗ | / | 0.850 (0.003) | 0.057 (0.034) | 0.117 |
| | ✓ | 0.10 | 0.850 (0.003) | **0.036** (0.025) | **0.083** |
| **FairFed** | ✗ | / | 0.842 (0.003) | 0.069 (0.034) | 0.118 |
| | ✓ | 0.10 | 0.841 (0.003) | **0.037** (0.026) | **0.081** |

Table 6 shows the result on adult with parameter for Dirichlet distribution=10. Moreover, we present an analysis of the impact of parameter variations on the experimental results. We consider two parameters——the fairness constraint, $\alpha$, and the confidence coefficient, $\beta$, separately. Figure 3 and 4 shows the result on Adult dataset and Compas dataset, respectively.

Table 5: **Results with standard deviation on Compas.**

| Model | FedFaiREE | $\alpha$ | $\overline{ACC}$ | $\overline{|DEOO|}$ | $|DEOO|_{95}$ |
|---|---|---|---|---|---|
| | | | **Compas** | | |
| **FedAvg** | ✗ | / | 0.662 (0.011) | 0.126 (0.056) | 0.223 |
| | ✓ | 0.15 | 0.659 (0.010) | **0.051** (0.044) | **0.137** |
| **AFL** | ✗ | / | 0.643 (0.012) | 0.097 (0.050) | 0.170 |
| | ✓ | 0.15 | 0.641 (0.011) | **0.051** (0.033) | **0.108** |
| **FedFB** | ✗ | / | 0.642 (0.011) | 0.107 (0.043) | 0.174 |
| | ✓ | 0.15 | 0.641 (0.010) | **0.062** (0.040) | **0.125** |
| **FairFed** | ✗ | / | 0.648 (0.012) | 0.097 (0.047) | 0.166 |
| | ✓ | 0.15 | 0.645 (0.011) | **0.047** (0.036) | **0.114** |

Table 6: **Results on Adult with Parameter for Dirichlet distribution=10.**

| Model | FedFaiREE | $\alpha$ | $\overline{ACC}$ | $\overline{|DEOO|}$ | $|DEOO|_{95}$ |
|---|---|---|---|---|---|
| | | | **Adult** | | |
| **FedAvg** | ✗ | / | 0.844 (0.004) | 0.127 (0.032) | 0.184 |
| | ✓ | 0.10 | 0.843 (0.003) | **0.029** (0.027) | **0.091** |
| **AFL** | ✗ | / | 0.848 (0.004) | 0.098 (0.037) | 0.168 |
| | ✓ | 0.10 | 0.848 (0.004) | **0.033** (0.026) | **0.082** |
| **FedFB** | ✗ | / | 0.845 (0.003) | 0.057 (0.034) | 0.117 |
| | ✓ | 0.10 | 0.845 (0.003) | **0.036** (0.025) | **0.083** |
| **FairFed** | ✗ | / | 0.839 (0.004) | 0.081 (0.033) | 0.138 |
| | ✓ | 0.10 | 0.838 (0.004) | **0.027** (0.025) | **0.073** |

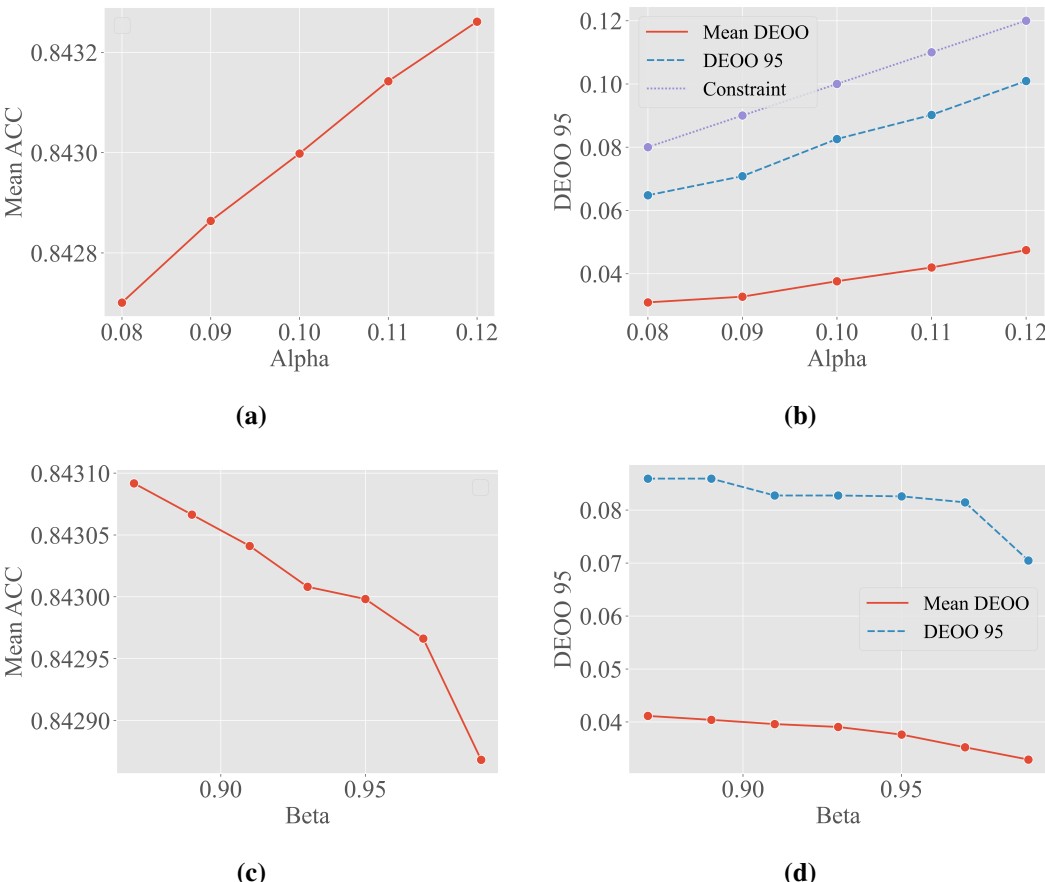

Figure 3: **The changes of accuracy, $\overline{|DEOO|}$ and $|DEOO|_{95}$ with respect to $\alpha$ and $\beta$ on Adult.** The other parameters of the experiment are consistent with those in Table 1.

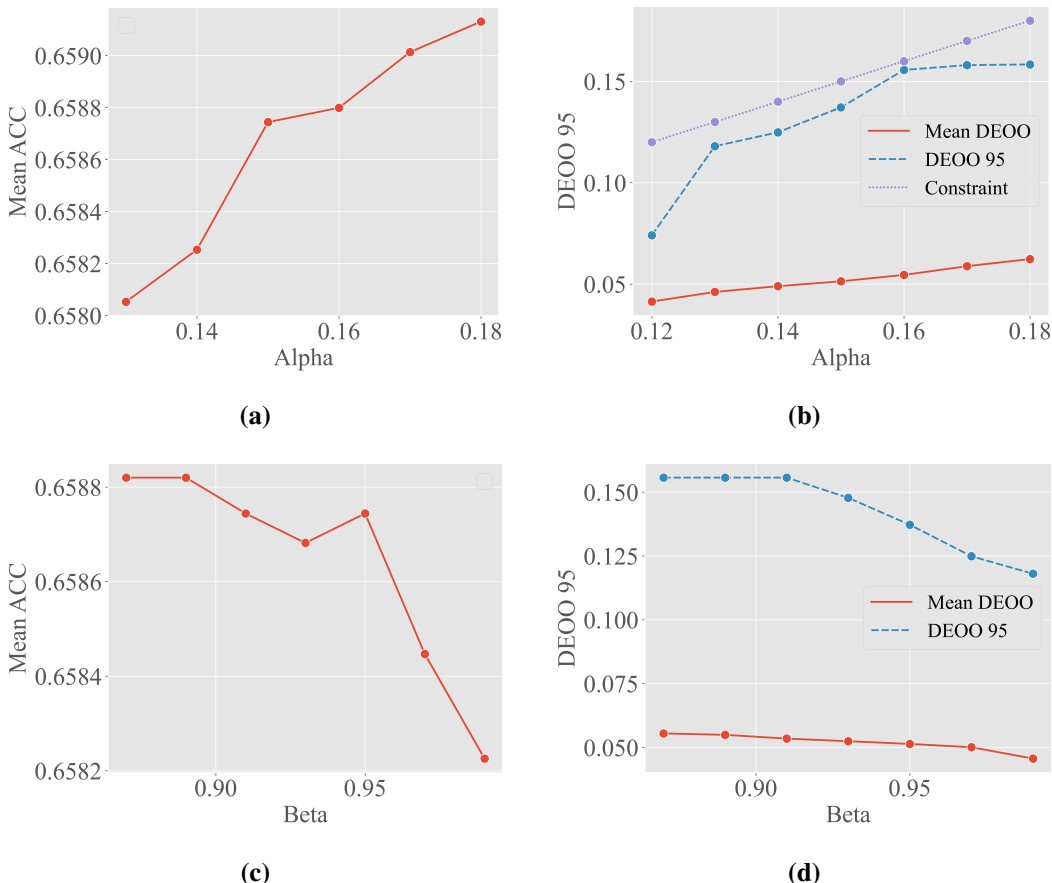

Figure 4: **The changes of accuracy, $\overline{|DEOO|}$ and $|DEOO|_{95}$ with respect to $\alpha$ and $\beta$ on Compas.** The other parameters of the experiment are consistent with those in Table 1.

### C.4 FURTHER RESULTS ON DEO

In this subsection, we conducted experiments using FedFaiREE for DEO, which is a specific algorithm under the FedFaiREE framework designed for DEO as mentioned in Section 5.2. The results are presented in Tables 7 and 8. It's worth noting that FedFaiREE for DEO exhibited favorable performance similar to FedFaiREE for DEOO, showing significant improvements in both DEOO and DPE indicators while maintaining relatively high accuracy.

### C.5 FURTHER EXPERIMENT RESULT ON OTHER DATASET

In this subsection, we use ACSIncome(Ding et al., 2021) to present the performance of FedFaiREE. ACSIncome dataset is constructed by the American Community Survey (ACS) Public Use Microdata Sample (PUMS), whose task is to predict whether a an individual's income is above $50k with attributes including age, employment type, education, martial status, etc. The sensitive attribute we consider is whether white people or not.

The ACSIncome dataset comprises 1,664,500 samples, and we consider each state representing a separate client (totaling 51 states). To evaluate the performance of our method in scenarios involving small sample sizes, we conducted random sampling, selecting 0.01 proportion of data from each state for a total of 100 iterations. Subsequently, we performed an 80-20 split on each selected subset, creating distinct training and test sets for analysis.

Table 7: **Results of FedFaiREE for DEO on Adult dataset.** We conducted 100 experimental repetitions for each model on both datasets and compared the accuracy and fairness indicators of different models. The "FedFaiREE" and "$\alpha$" columns indicate whether FedFaiREE was used or not. "$\overline{ACC}$", "$\overline{|DEOO|}$" and "$\overline{|DPE|}$" represent the averages of accuracy, DEOO (defined in Equation 1) and DPE (defined in Equation 27), respectively. "$|DEOO|_{95}$" and "$|DPE|_{95}$" represent the 95% quantile of DEOO and DPE since we set the confidence level of FedFaiREE to 95% in our experiments.

| Model | FedFaiREE | $\alpha$ | $\overline{ACC}$ | $\overline{|DEOO|}$ | $|DEOO|_{95}$ | $\overline{|DPE|}$ | $|DPE|_{95}$ |
|---|---|---|---|---|---|---|---|
| | | | | **Adult** | | | |
| **FedAvg** | No | / | 0.844 (0.003) | 0.131 (0.030) | 0.178 | 0.088 (0.005) | 0.097 |
| | Yes | 0.10 | 0.843 (0.003) | **0.037** (0.025) | **0.082** | **0.064** (0.007) | **0.075** |
| **AFL** | No | / | 0.848 (0.004) | 0.101 (0.040) | 0.169 | 0.095 (0.012) | 0.118 |
| | Yes | 0.10 | 0.847 (0.004) | **0.031** (0.026) | **0.079** | **0.069** (0.011) | **0.079** |
| **FedFB** | No | / | 0.850 (0.003) | 0.057 (0.034) | 0.117 | 0.066 (0.007) | 0.077 |
| | Yes | 0.10 | 0.850 (0.003) | **0.036** (0.025) | **0.083** | **0.061** (0.006) | **0.070** |
| **FairFed** | No | / | 0.842 (0.003) | 0.069 (0.034) | 0.118 | 0.072 (0.006) | 0.083 |
| | Yes | 0.10 | 0.841 (0.003) | **0.037** (0.026) | **0.081** | **0.063** (0.006) | **0.071** |

Table 8: **Results of FedFaiREE for DEO on Compas dataset.**

| Model | FedFaiREE | $\alpha$ | $\overline{ACC}$ | $\overline{|DEOO|}$ | $|DEOO|_{95}$ | $\overline{|DPE|}$ | $|DPE|_{95}$ |
|---|---|---|---|---|---|---|---|
| | | | | **Compas** | | | |
| **FedAvg** | ✗ | / | 0.662 (0.011) | 0.126 (0.056) | 0.223 | 0.083 (0.032) | 0.136 |
| | ✓ | 0.15 | 0.652 (0.036) | **0.049** (0.045) | **0.137** | **0.028** (0.024) | **0.072** |
| **AFL** | ✗ | / | 0.643 (0.012) | 0.097 (0.050) | 0.170 | 0.089 (0.035) | 0.150 |
| | ✓ | 0.15 | 0.641 (0.011) | **0.049** (0.033) | **0.108** | **0.034** (0.025) | **0.074** |
| **FedFB** | ✗ | / | 0.642 (0.011) | 0.107 (0.043) | 0.174 | 0.066 (0.028) | 0.112 |
| | ✓ | 0.15 | 0.642 (0.010) | **0.062** (0.040) | **0.125** | **0.036** (0.024) | **0.081** |
| **FairFed** | ✗ | / | 0.648 (0.011) | 0.097 (0.047) | 0.166 | 0.087 (0.036) | 0.148 |
| | ✓ | 0.15 | 0.642 (0.029) | **0.047** (0.036) | **0.114** | **0.037** (0.028) | **0.085** |

Table 9: **Results of FedFaiREE on ACSIncome.**

| Model | FedFaiREE | $\alpha$ | $\overline{ACC}$ | $\overline{|DEOO|}$ | $|DEOO|_{95}$ |
|---|---|---|---|---|---|
| | | | | **ACSIncome** | |
| **FedAvg** | ✗ | / | 0.788 (0.008) | 0.108 (0.045) | 0.185 |
| | ✓ | 0.1 | 0.786 (0.008) | **0.040** (0.030) | **0.089** |
| **AFL** | ✗ | / | 0.774 (0.009) | 0.093 (0.043) | 0.158 |
| | ✓ | 0.1 | 0.771 (0.013) | **0.043** (0.031) | **0.101** |
| **FedFB** | ✗ | / | 0.771 (0.008) | 0.086 (0.049) | 0.172 |
| | ✓ | 0.1 | 0.770 (0.008) | **0.049** (0.039) | **0.102** |
| **FairFed** | ✗ | / | 0.774 (0.009) | 0.087 (0.046) | 0.160 |
| | ✓ | 0.1 | 0.772 (0.008) | **0.043** (0.035) | **0.100** |

# D EXTENSION TO MULTI-GROUPS AND MULTI-LABELS FAIRNESS

## D.1 EXTENSION TO MULTI-GROUPS

**Definition D.1.** *(Equality of Opportunity, Multiple Groups) A classifier satisfies Equality of Opportunity if it satisfies the same true positive rate among protected groups:*

$$\mathbb{P}_{X|A=0,Y=1}(\widehat{Y}=1) = \mathbb{P}_{X|A=a,Y=1}(\widehat{Y}=1),$$

*where $a$ belongs to a protected class $\mathcal{A} = \{1, \cdots, A_0\}$*

Similar to $DEOO$, we define metric for Equality of Opportunity under Multiple Groups as:

$$DEOOM = \max_a\{|\mathbb{P}_{X|A=a,Y=1}(\widehat{Y}=1) - \mathbb{P}_{X|A=0,Y=1}(\widehat{Y}=1)|\}$$

Therefore, inspired by Proposition 3.2, we have

**Proposition D.1.** *Under Assumption 3.1, for $a \in \{0, 1, \cdots, A_0\}$, consider $k^{1,a} \in \{1, \ldots, n^{1,a}\}$, the corresponding $\hat{k}_i^{1,a}$ for $i \in [S]$ which are $\varepsilon$-approximate ranks and the score-based classifier $\phi(x,a) = \mathbb{1}\{f(x,a) > t_{(k^{1,a})}^{1,a}\}$. Define*

$$
\begin{aligned}
h_{y,a}^* = &\mathbb{P}\left(\sum_{i=1}^S \pi_i^{y,a} Q\left(M_i^{1,a}, n_i^{y,a} + 1 - M_i^{1,a}\right) - \sum_{i=1}^S \pi_i^{y,0} Q\left(m_i^{1,0}, n_i^{y,0} + 1 - m_i^{1,0}\right) \geq \alpha\right) \\
&+ \mathbb{P}\left(\sum_{i=1}^S \pi_i^{y,0} Q\left(M_i^{1,0}, n_i^{y,0} + 1 - M_i^{1,0}\right) - \sum_{i=1}^S \pi_i^{y,a} Q\left(m_i^{1,a}, n_i^{y,a} + 1 - m_i^{1,a}\right) \geq \alpha\right)
\end{aligned}
$$

*Then we have:*

$$\mathbb{P}(|DEOOM(\phi)| > \alpha) \leq \sum_{a=1}^{A_0} h_{1,a}^* \tag{39}$$

*where $\pi_i^{1,a}$, $\pi_i^{1,0}$ are similarly defined as in Proposition 3.2. $M_i^{1,a} = max\left(\lceil \hat{k}_i^{1,a} + \varepsilon n_i^{1,a}\rceil, n_i^{1,a}+1\right)$, $m_i^{1,a} = min\left(\lceil \hat{k}_i^{1,a} - \varepsilon n_i^{1,a}\rceil, 0\right)$, $M_i^{1,0}$ and $m_i^{1,0}$ are similarly defined. $Q(\alpha,\beta)$ are independent random variables and $Q(\alpha,\beta) \sim Beta(\alpha,\beta)$. Especially, we define $Q(0,\beta) = 0$ and $Q(\alpha,0) = 1$ for $\alpha, \beta \neq 0$.*

Proposition D.1 can be regarded as a direct corollary of Proposition 3.2. Moveover, similar to Proposition 3.3, we have

**Proposition D.2.** *Under Assumption 3.1, the misclassification error can be estimated by*

$$\hat{\mathbb{P}}\left(\hat{\phi}(x,a) \neq Y\right) = \sum_{i=1}^S \left[\pi_i \sum_{a=0}^{A_0} \left(\frac{\hat{k}_i^{1,a}+0.5}{n_i^{1,a}+1} p_a^i p_{Y,a}^i + \frac{n_i^{0,a}+0.5-\hat{k}_i^{0,a}}{n_i^{0,a}+1} p_a^i q_{Y,a}^i\right)\right] \tag{40}$$

*Further, the discrepancy between empirical error and true error is upper bounded by the following:*

$$\left|\mathbb{P}\left(\hat{\phi}(x,a) \neq Y\right) - \hat{\mathbb{P}}\left(\hat{\phi}(x,a) \neq Y\right)\right| \leq \theta, \tag{41}$$

*where $\theta = \sum_{i=1}^S \left[\pi_i \sum_{a=0}^{A_0} \left(e_i^{0,a} p_a^i q_{Y,a}^i + e_i^{1,a} p_1^i q_{Y,a}^i\right)\right]$, $e_i^{y,a} = \frac{2\lfloor \varepsilon n_i^{y,a}\rfloor+1}{2(n_i^{y,a}+1)}$*

**Theorem D.2.** *Under Assumption 3.1 and 4.1, given $\alpha' < \alpha$. Suppose $\hat{\phi}$ is the final output of FedFaiREE, we have:*

*(1) $|DEOOM(\hat{\phi})| < \alpha$ with probability $(1-\delta)^N$, where $N$ is the size of the candidate set.*

*(2) Suppose the density distribution functions of $f^*$ under $A = a, Y = 1$ are continuous. When the input classifier $f$ satisfies $|f(x,a) - f^*(x,a)| \leq \epsilon_0$, for any $\epsilon > 0$ such that $F_{(+)}^*(\epsilon + \gamma\varepsilon) \leq \frac{\alpha-\alpha'}{2} - F_{(+)}^*(2\epsilon_0)$, we have*

$$\mathbb{P}(\hat{\phi}(x,a) \neq Y) - \mathbb{P}(\phi_{\alpha'}^*(x,a) \neq Y) \leq 2F_{(+)}^*(2\epsilon_0) + 2F_{(+)}^*(\epsilon + \gamma\varepsilon) + 2\theta + O(\epsilon) \tag{42}$$

*with probability $1 - 4\sum_{a=0}^{A_0}\sum_{i=1}^S e^{-2n_i^{0,a}\epsilon^2} - \sum_{a=0}^{A_0}\prod_{i=1}^S \left(1 - F_{i(-)}^{1,a}(2\epsilon)\right)^{n_i^{1,a}} - \delta$, where $\delta = \sum_{a=0}^{A_0} \delta^{1,a}(n^{1,a})$, $\theta$ is defined in Proposition D.2 and the definition of $F_{(+)}$ and $F_{(-)}$ are shown in Lemma A.4*

---

**Algorithm 7** FedFaiREE for Multi-Groups

---

**Input:** Train dataset $D_i = D_i^{0,0} \cup D_i^{0,1} \cup D_i^{1,0} \cup D_i^{1,1}$; pre-trained classifier $\phi_0$ with function f; fairness constraint parameter $\alpha$ ; Confidence level parameter $\beta$; Weights of different clients $\pi$

**Output:** classifier $\hat{\phi}(x,a) = \mathbf{1}\{f(x,a) > t_{(k^{1,a})}^{1,a}\}$

**Client Side:**

**for** i=1,2,...,S **do**

    Score on train data points in $D_i$ and get $T_i^{y,a} = \{t_{i,1}^{y,a}, t_{i,2}^{y,a}, \cdots, t_{i,n_i^{y,a}}^{y,a}\}$

    Sort $T_i^{y,a}$

    Calculate q-digest of $T_i^{y,a}$ on client $i$

    Update digest to server

**end for**

**Server Side:**

Construct $K$ by $K = \{(\boldsymbol{k}^{1,0}, \boldsymbol{k}^{1,1}, \cdots, \boldsymbol{k}^{1,A_0}) | L < 1 - \beta\}$, where L is defined by the right-hand side of Inequality 39

Select optimal $(\boldsymbol{k}^{1,0}, \boldsymbol{k}^{1,1}, \cdots, \boldsymbol{k}^{1,A_0})$ by minimizing equation 40 using estimated values $\hat{p}_a^i$ and $\hat{p}_{Y,a}^i$

---

### D.2 EXTENSION TO MULTI-LABELS

**Definition D.3.** *(Equality of Opportunity, Multiple labels(Liu et al., 2023)) A classifier satisfies Equality of Opportunity if it satisfies :*

$$\hat{\boldsymbol{Y}} \perp A \mid \boldsymbol{Y} = \boldsymbol{y}_{adv},$$

*where $\boldsymbol{Y} \in \{0,1\}^m$ and $\boldsymbol{y}_{adv}$ denotes some advantaged label where only favorable outcomes.*

**Definition D.4.** *(Multi-label Score-based Classifier) A Multi-label score-based classifier is an element-wise indication function, where the j-th component of $\hat{\boldsymbol{Y}}$ satisfies $\hat{Y}_j = \phi_j(x,a) = \mathbb{1}\{f_j(x,a) > c_j\}$ for a measurable score function $f : \mathcal{X} \times \{0,1\} \to [0,1]$ and a constant threshold $c_j > 0$.*

Considering relaxing the aforementioned Equality of Opportunity constraint, we introduce a fairness indicator as follow:

$$DEOOM_{\boldsymbol{y}}(\phi) = \left|\mathbb{P}[\hat{\boldsymbol{Y}} = \boldsymbol{y} \mid A = 0, \boldsymbol{Y} = \boldsymbol{Y}_{adv}] - \mathbf{P}[\hat{\boldsymbol{Y}} = \boldsymbol{y} \mid A = 1, \boldsymbol{Y} = \boldsymbol{Y}_{adv}]\right|,$$

where $\boldsymbol{y}$ can be considered as either certain advantageous labels or as a collection of advantageous labels (at this point, '=' is replaced by '$\in$').

Additionally, we consider an iterative Q-digest approach. At each client, our process involves constructing a Q-digest initially for the first component of the score $f(x)$. Subsequently, at each leaf node, we include a Q-digest for the second component of score $f(x)$ associated with the leaf node's first component. Repeating this procedure iteratively allows us to generate a sketch for the multidimensional score function $f(x)$. Assuming the parameter is appropriately set to achieve an $\varepsilon_j$-approximate quantile and rank for the $j$-th component, we arrive at the following result.

**Proposition D.3.** *Under Assumption 3.1, for $a \in \{0,1\}$, consider $\boldsymbol{q}^{\boldsymbol{y}_{adv},a} = (q_1^{\boldsymbol{y}_{adv},a}, q_2^{\boldsymbol{y}_{adv},a}, ..., q_m^{\boldsymbol{y}_{adv},a}) \in [0,1]^m$, $n_{i,(j)}^{\boldsymbol{y}_{adv},a}$ is the estimation of $|N_{i,(j)}^{\boldsymbol{y}_{adv},a}|$, $N_{i,(j)}^{\boldsymbol{y}_{adv},a} = \{f_j(x) \mid x \text{ belongs to Client } i, Y = \boldsymbol{y}_{adv}, A = a, (f_l(x) - t_l)y_l^* \geq 0, l = 1, \cdots, j-1\}$ and $t_j^{\boldsymbol{y}_{adv}}$ is estimation of $q_j$ quantile of $N_{*,(j)}^{\boldsymbol{y}_{adv},a}$ (the union of $N_{i,(j)}^{\boldsymbol{y}_{adv},a}$), where estimations with subscript $(j)$ are $\varepsilon$-approximate ranks and quantiles, $\hat{k}_{i,(j)}^{\boldsymbol{y}_{adv},a}$ represent the estimation local rank of $t_j^{\boldsymbol{y}_{adv}}$ in $N_{i,(j)}^{\boldsymbol{y}_{adv},a}$, the score-based classifier $\phi(x,a) = \mathbb{1}\{f(x,a) > t_j^{\boldsymbol{y}_{adv},a}\}$. Define*

$$h_{\boldsymbol{y}_{adv},a} = \mathbb{P}\left(\sum_{i=1}^{S} \pi_i^{\boldsymbol{y}_{adv},a} \prod_{j=1}^{m} g_j \left(Q\left(u_{i,(j)}^{\boldsymbol{y}_{adv},a}, (1 + (1 - 2y_j^*)\varepsilon_{j-1})n_{i,(j)}^{\boldsymbol{y}_{adv},a} + 1 - u_{i,(j)}^{\boldsymbol{y}_{adv},a}\right)\right)\right.$$

$$\left. - \sum_{i=1}^{S} \pi_i^{\boldsymbol{y}_{adv},1-a} \prod_{j=1}^{m} g_j \left(Q\left(v_{i,(j)}^{\boldsymbol{y}_{adv},1-a}, (1 + (2y_j^* - 1)\varepsilon_{j-1})n_{i,(j)}^{\boldsymbol{y}_{adv},1-a} + 1 - v_{i,(j)}^{\boldsymbol{y}_{adv},1-a}\right)\right) \geq \alpha\right),$$

*Then we have:*

$$\mathbb{P}(|DEOOM_{\boldsymbol{y}^*}(\phi)| > \alpha) \leq h_{\boldsymbol{y}_{adv},0} + h_{\boldsymbol{y}_{adv},1}, \tag{43}$$

*where $\pi_i^{\boldsymbol{y}_{adv},a}$ is similarly defined as in Proposition 3.1, $g_j(Q) = (1 - 2y_j^*)Q + y_j^*, u_{i,(j)}^{\boldsymbol{y}_{adv},a} = y_j^* m_{i,(j)}^{\boldsymbol{y}_{adv},a} + (1-y_j^*)M_{i,(j)}^{\boldsymbol{y}_{adv},a}, v_{i,(j)}^{\boldsymbol{y}_{adv},a} = y_j^* M_{i,(j)}^{\boldsymbol{y}_{adv},a} + (1-y_j^*)m_{i,(j)}^{\boldsymbol{y}_{adv},a}, M_{i,(j)}^{\boldsymbol{y}_{adv},a} = max(\lceil \hat{k}_{i,(j)}^{\boldsymbol{y}_{adv},a} + \varepsilon_j n_{i,(j)}^{\boldsymbol{y}_{adv},a} \rceil, n_{i,(j)}^{\boldsymbol{y}_{adv},a} + 1), m_{i,(j)}^{\boldsymbol{y}_{adv},a} = min(\lceil \hat{k}_{i,(j)}^{\boldsymbol{y}_{adv},a} - \varepsilon_j n_{i,(j)}^{\boldsymbol{y}_{adv},a} \rceil, 0)$, and $Q(\alpha,\beta)$ are independent random variables and $Q(\alpha,\beta) \sim Beta(\alpha,\beta)$. Especially, we define $Q(0,\beta) = 0$ and $Q(\alpha,0) = 1$ for $\alpha, \beta \neq 0$.*

Similar to Proposition 3.2, the proposition above can be proved using Lemma A.1 and conditional probability. It is important to note that $\boldsymbol{y}$ and $\boldsymbol{y}_{adv}$ are not necessarily single labels; they can also represent a set of labels with constraints on specific components where values are restricted to 0 or 1 (for $j$ where $y_j^*$ does not have constraint, $t_j$ is set to 0.5, and it is excluded from the construction of $N$ and calculation of $h$). And similarly, the selection can be conducted by minimizing empirical misclassification error.

Considering a high-dimensional extension of Lemma A.4, we have

**Lemma D.5.** *For a distribution $F$ with a continuous density function, suppose $q(x)$ denotes the probability of $X \preceq x$ where $X$ is a random variable under $F$, then for $y \preceq x$, we have $F_{(-)}(\|x - y\|_2) \leq q(x) - q(y) \leq F_{(+)}(\|x - y\|_2)$, where $F_{(-)}(x)$ and $F_{(+)}(x)$ are two monotonically increasing functions, $F_{(-)}(\epsilon) > 0, F_{(+)}(\epsilon) > 0$ for any $\epsilon > 0$ and $\lim_{\epsilon \to 0} F_{(-)}(\epsilon) = \lim_{\epsilon \to 0} F_{(+)}(\epsilon) = 0$.*

Therefore, similarly, we have

**Theorem D.6.** *Under Assumption 3.1 and 4.1, given $\alpha' < \alpha$. Suppose $\hat{\phi}$ is the final output of FedFaiREE, we have:*

*(1) $|DEOOM_{\boldsymbol{y}^*}(\hat{\phi})| < \alpha$ with probability $(1 - \delta)^N$, where $N$ is the size of the candidate set.*

*(2) Suppose the density distribution functions of $f^*$ under $A = a, Y = 1$ are continuous. When the input classifier $f$ satisfies $\|f(x,a) - f^*(x,a)\|_2 \leq \epsilon_0$, for any $\epsilon > 0$ such that $M_{(+)}^*(\epsilon + \gamma\varepsilon) \leq \frac{\alpha - \alpha'}{2m} - M_{(+)}^*(2\epsilon_0)$, we have*

$$\mathbb{P}(\hat{\phi}(x,a) \neq Y) - \mathbb{P}(\phi_{\alpha'}^*(x,a) \neq Y) \leq 2mM_{(+)}^*(2\epsilon_0) + 2mM_{(+)}^*(\epsilon + \gamma\varepsilon_m) + 2\theta + O(\epsilon) \tag{44}$$

*with probability $1 - (2^{m+1} + 2)\sum_{a=0}^1 \sum_{i=1}^S e^{-2n_i^{0,a}\epsilon^2} - \sum_{a=0}^1 \prod_{i=1}^S \left(1 - F_{i(-)}^{\boldsymbol{y}_{adv},a}(2\epsilon)\right)^{n_i^{\boldsymbol{y}_{adv},a}} - \delta$, where $\delta = \sum_{a=0}^1 \delta^{\boldsymbol{y}_{adv},a}(n^{\boldsymbol{y}_{adv},a})$, $\theta = \sum_{i=1}^S \left[\pi_i \sum_{a=0}^1 \sum_{\boldsymbol{y}} e_i^{\boldsymbol{y},a} p_a^i p_{\boldsymbol{y},a}^i\right]$, $e_i^{\boldsymbol{y},a} = \frac{2\lfloor \varepsilon_m n_i^{\boldsymbol{y},a}\rfloor + 1}{2(n_i^{\boldsymbol{y},a} + 1)}$, $M_{(+)}^*$ corresponds to the maximum of $F_{(+)}$ associated with $f_j^*$, and the definition of $F_{(+)}$ and $F_{(-)}$ are shown in Lemma D.5.*

## E    COMPARISON TO FAIREE(LI ET AL., 2022) AND OTHER RELATED WORKS

Regarding the differences between FedFaiREE and FaiREE, several pivotal distinctions become evident. Primarily, FedFaiREE demonstrates superior adaptability for practical applications. Notably, it incorporates mechanisms to handle label shift scenarios, ensuring model robustness within such distributions, as elucidated in Section 5.1. Furthermore, it's worth noting that FedFaiREE extends considerations to encompass multiple sensitive groups and multiple labels, aligning more closely with practical real-world application scenarios, as discussed in Appendix D.

Another critical difference lies in the setting: FaiREE operates in a centralized environment, assuming homogeneous data across all clients. In contrast, FedFaiREE is expressly tailored for decentralized settings, acknowledging client heterogeneity and effectively addressing the challenges stemming from diverse data distributions and sizes across clients. This tailored approach significantly enhances its adaptability and robustness across various scenarios.

Lastly, while FaiREE relies on specific centralized quantile estimation methods, FedFaiREE adopts approximate quantiles. This adaptation not only facilitates adaptation to distributed data but also fortifies the method's robustness and adaptability.

### E.1 COMPARISON TO OTHER RELATED WORKS

Differences between FedFaiREE and other fair federated learning methods lie in their approach to addressing fairness concerns. Many methods, akin to this paper, extend the principles of centralized machine learning to decentralized settings, such as FedFB(Zeng et al., 2021), FedMinMax(Papadaki et al., 2022), PFFL(Hu et al., 2022), and others. These methods primarily focus on introducing fairness penalties in the objective functions and incorporate client reweighting schemes and terms (in objective functions) reweighting schemes that consider global or local fairness. The key divergence between our approach and these methods is that the latter typically converge and provide fairness guarantees only in large-sample scenarios, lacking assurances for fairness in small-sample situations, especially under distribution-free assumptions. Empirical results from Table 1 in this paper demonstrate that compared to FedFaiREE, methods like FedFB, FairFed are not as effective in controlling fairness in small-sample scenarios. Furthermore, as these methods are predominantly in-processing techniques, while FedFaiREE falls under post-processing methods, there is a potential for further integration to achieve improved fairness guarantees as shown in our experiments. Moreover, another significant characteristic of FedFaiREE is its capability to adjust the trade-off between fairness and accuracy according to specific fairness constraints. This control capacity has been demonstrated in numerous experiments, showcasing an ability that other methods lack.

