# OpenReview forum: "Distribution-Free Fair Federated Learning with Small Samples"
_ICLR.cc/2024/Conference — Submitted to ICLR 2024_

### Official Review · Reviewer_6gnc · 2023-10-22

**Soundness:** 3 good
**Presentation:** 3 good
**Contribution:** 2 fair
**Rating:** 5
**Confidence:** 3

**Summary:**

The paper introduces a novel algorithm called FedFaiREE, designed to tackle fairness concerns in federated learning. Existing fairness algorithms primarily cater to centralized data environments, relying on large samples and distributional assumptions. This paper addresses the pressing need for fairness techniques tailored to decentralized systems with finite samples and distribution-free guarantees. FedFaiREE is specifically developed for distribution-free fair learning in decentralized settings with small samples. The algorithm considers the unique challenges posed by decentralized environments, including client heterogeneity, communication costs, and small sample sizes commonly encountered in practical scenarios. The paper offers rigorous theoretical guarantees for both fairness and accuracy, and the experimental results provide strong empirical validation of these theoretical claims.

**Strengths:**

The strengths of FedFaiREE are as follows:

1. **Effective Fairness in Challenging Conditions:** FedFaiREE offers a simple yet highly effective approach for ensuring fairness in scenarios with limited samples and distribution-free conditions. This is particularly important in real-world applications where such conditions are prevalent.

2. **Theoretical Fairness Guarantees:** The paper provides theoretical guarantees that FedFaiREE can achieve nearly optimal fairness when the input prediction function is appropriate. This adds a level of confidence in the algorithm's ability to deliver on its fairness objectives.

**Weaknesses:**

My primary concern regarding this paper centers on its contribution when compared with previous work, particularly the FaiREE algorithm designed for centralized learning. Based on my interpretation, the overall process appears quite similar, with the primary distinction being the distributed aggregation process. I would greatly value it if the authors could delve deeper into the algorithmic variances and the technical challenges associated with the algorithm's design and theoretical analysis. This deeper exploration would enhance the paper's clarity and help readers better understand the specific advancements and innovations brought about by FedFaiREE in relation to its predecessor, FaiREE.

**Questions:**

see above

---

> ### Author Response · Authors · 2023-11-22
> **Response to Reviewer 6gnc**
>
> Thanks for your review and helping us improve our paper.
>
> >W: My primary concern regarding this paper centers on its contribution when compared with previous work, particularly the FaiREE algorithm designed for centralized learning. Based on my interpretation, the overall process appears quite similar, with the primary distinction being the distributed aggregation process. I would greatly value it if the authors could delve deeper into the algorithmic variances and the technical challenges associated with the algorithm's design and theoretical analysis. This deeper exploration would enhance the paper's clarity and help readers better understand the specific advancements and innovations brought about by FedFaiREE in relation to its predecessor, FaiREE.
>
> - A1: For Weakness and Question, regarding the differences between FedFaiREE and FaiREE, in light of your comment, we have incorporated the following discussion highlighting the differences between FedFaiREE and FaiREE:
>
> “Regarding the differences between FedFaiREE and FaiREE, several pivotal distinctions become evident. Primarily, FedFaiREE demonstrates superior adaptability for practical applications. Notably, it incorporates mechanisms to handle label shift scenarios, ensuring model robustness within such distributions, as elucidated in Section 5.1. Furthermore, it's worth noting that FedFaiREE extends considerations to encompass multiple sensitive groups and multiple labels, aligning more closely with practical real-world application scenarios, as discussed in Appendix D.
>
> Another critical difference lies in the setting: FaiREE operates in a centralized environment, assuming homogeneous data across all clients. In contrast, FedFaiREE is expressly tailored for decentralized settings, acknowledging client heterogeneity and effectively addressing the challenges stemming from diverse data distributions and sizes across clients. This tailored approach significantly enhances its adaptability and robustness across various scenarios.
>
> Lastly, while FaiREE relies on specific centralized quantile estimation methods, FedFaiREE adopts approximate quantiles. This adaptation not only facilitates adaptation to distributed data but also fortifies the method's robustness and adaptability.”

---

### Official Review · Reviewer_7R45 · 2023-10-27

**Soundness:** 3 good
**Presentation:** 3 good
**Contribution:** 3 good
**Rating:** 8
**Confidence:** 3

**Summary:**

This paper proposes a versatile post-processing method that can be used together with other pre-processing and in-processing techniques to ensure fairness in federated learning. The method is distribution-free and only requires small samples and communication costs. Under a binary label prediction task, the authors derived theoretical guarantees for both model fairness and performance under Equality of Opportunity, after which was further extended to a distribution shift setting and an extended Equalized Odds fairness notion. Empirical experiments on Adults and Compas datasets were also carried out to demonstrate the superior performance and fairness of the proposed FedFaiREE method when compared to other baselines.

**Strengths:**

1. The paper is clearly written and has strong motivation paragraphs with well-categorized related works.
2. The authors further study estimation and approximation methods for better adoption in practice.
3. The theoretical results and analyses are sound. The guarantees derived are further extended to a setting with test distribution shifts and also a stronger extended notion of fairness.
4. The method proposed is versatile since it can be used in combination with other pre-processing and in-processing techniques to ensure better fairness.

**Weaknesses:**

1. The setting of the paper may be restrictive since it only applies to tabular data with binary prediction labels. Nevertheless, simpler settings might be needed for the ease of analysis.
2. The empirical validation of the method and performance is not comprehensive enough.

Some other details are given below in the Questions section.

**Questions:**

1. Help me understand the theoretical results: How tight is the bound that is derived in (7)? If the optimal classifier is indeed unfair, how can you still achieve an optimal misclassification error with the DEOO constraint?
2. How difficult is the extension to multiple labels? The current empirical validation section still appears not as convincing due to the simplistic setting and well-behaved binary label prediction datasets used. A larger-scale setting also supports the necessity of federated learning.
3. Why is the same $\alpha$ used for all experiments on a dataset (i.e., 0.1 for Adult and 0.15 for Compas)? How should this alpha be set in practice?
4. What is the trend of accuracy (ACC) when we shrink $\alpha$? What is a good point to stop (for $\alpha$) in order to balance accuracy and fairness? What is the lowest value of $\alpha$ we can go?

---

> ### Author Response · Authors · 2023-11-22
> **Response to Reviewer 7R45**
>
> Thank you for your review and for contributing to the improvement of our paper.
> >Q1: Help me understand the theoretical results: How tight is the bound that is derived in (7)? If the optimal classifier is indeed unfair, how can you still achieve an optimal misclassification error with the DEOO constraint?
>
> - A1: In our paper, the fairness-constrained Bayes-optimal classifier is defined as the classifier with the highest accuracy under the desired DEOO constraints.  Notably, the construction of the fairness-constrained Bayes-optimal classifier involves employing group-wise shifts derived from the unconstrained Bayes-optimal classifier[1]. This approach serves as a motivation behind the development of the FedFaiREE pipeline. The major results in equation (7) encompass two aspects: one extends from empirical misclassification to global misclassification, and the other progresses from the optimal global misclassification within the candidate set to the Bayes optimal classifier. The former involves error bounds concerning approximate quantiles, the use of Hoeffding's inequality for empirical estimation of $p_a$ and $p_{Y,a}$ and embedding these empirical estimations into the error. In this part, only the embedding could potentially be further optimized, with an upper bound space of at most O($\epsilon$). The bounds in the latter aspect mainly involve estimates of distribution function characteristics (Lemma A.4).
>
> >Q2: How difficult is the extension to multiple labels? The current empirical validation section still appears not as convincing due to the simplistic setting and well-behaved binary label prediction datasets used. A larger-scale setting also supports the necessity of federated learning.
> - A2: Thank you for you suggestion, we added consideration about multiple labels in Appendix D.2. It is possible to guarantee fairness in such scenario. Considering computing resource limitation, we would add larger scale experiments later.
> In the case of multiple sensitive groups, referring to the DEOO concept (defined in equation (1) in the paper), we primarily consider the maximum value of DEOO between these protected groups and the unprotected residual portions to control the overall relaxation of Equality of Opportunity (see Definition D.1 for the multi-group version). At this point, similar to previous notions, we introduce fairness control by simultaneously managing multiple DEOOs and ensuring accuracy by incorporating considerations of $p_a$ and $p_{Y,a}$ across multiple groups.
>
> For the multi-label situation, we are inspired by [6] and consider the scenario where labels are m-dimensional vectors, where each component is binary. The results in Sections 3 and 4 of the article can be viewed as the case when m=1. In extending the one-dimensional results to higher dimensions, the main differences are as follows:
>
> 1. The different dimensions of labels may not be independent, hence, we introduce conditional distribution to address this. We estimate the second dimension based on the first dimension, and the third dimension based on both the first and second dimensions. This process is repeated for the remaining dimensions.
>
> 2. Q-digest cannot handle estimation of multi-dimensional distributions; therefore, we consider an iterative version of q-digest. First, we construct a q-digest based on the first dimension, then on the leaf nodes, we build q-digest based on the second dimension (the estimation error of quantiles in different dimensions differs).
>
>
> 3. In estimating accuracy, when extending from one dimension to higher dimensions, a closed interval in one dimension may no longer be a closed interval. Therefore, we have to consider its projection onto a closed interval in one dimension among the m dimensions.
>
> >Q3: Why is the same $\alpha$ used for all experiments on a dataset (i.e., 0.1 for Adult and 0.15 for Compas)? How should this alpha be set in practice?
> >
> >Q4: What is the trend of accuracy (ACC) when we shrink $\alpha$ What is a good point to stop (for $\alpha$) in order to balance accuracy and fairness? What is the lowest value of $\alpha$ we can go?
>
> - A3: For Questions 3 and 4, we appreciate your comment, and in response, we have included additional experiments featuring different fairness constraint parameters $\alpha$ and various confidence levels $\beta$ in Appendix C.3. In practice, setting $\alpha$ should align with the fairness requirement, but enforcing a strict fairness constraint might lead to a substantial loss in accuracy (consider choosing both $k_0$ and $k_1$ as 1). For instance, in the experiment on Compas, setting $\alpha$ to 0.13 yields an accuracy of approximately 66%, and it increases marginally as $\alpha$ grows. However, with $\alpha$ set to 0.12, the accuracy drops to below 60%.
>
> We value your feedback and have made the necessary additions and adjustments accordingly.
>
> [1] Zeng X, Dobriban E, Cheng G. Bayes-optimal classifiers under group fairness[J]. arXiv preprint, 2022.

---

> > ### Comment · Reviewer_7R45 · 2023-11-23
> >
> > I would like to thank the authors for the clarifications and the additional experiments. My concerns are resolved, please add the clarifications to the revised paper where appropriate. I would be happy to see a practical guideline for setting $\alpha$ in the revised paper, too. I am happy to keep my current score.

---

### Official Review · Reviewer_1atA · 2023-11-07

**Soundness:** 2 fair
**Presentation:** 1 poor
**Contribution:** 2 fair
**Rating:** 5
**Confidence:** 2

**Summary:**

This paper proposes a distribution-free post-processing approach to impose (approximate) equality of opportunity on a binary classifier with binary sensitive labels. The classifier is built as a sensitive-label-dependent threshold over a sensitive-label-dependent scoring function. This calibration procedure has finite sample guarantees.

**Strengths:**

The calibration procedure is distribution free and has finite sample guarantees. As a post-processing approach, it produces a simple pair of thresholds. To alleviate the communication costs of computing the quantile distribution across all clients, an efficient distributed quantile algorithm (Q-digest) is used.

**Weaknesses:**

Though conceptually simple, I found the presentation and the notation to be exceedingly hard to follow. I am also unsure on why, exactly, is there a need to maintain per-client score rankings other than to update the per-client sketches prior to aggregation.

Since the distributed quantile learning algorithm is not a contribution of this paper, and the notation is hard to follow, it is hard to evaluate the contribution and the insight of this work.

**Questions:**

what exactly is the use of the per-client score sorting?

---

> ### Author Response · Authors · 2023-11-22
> **Response to Reviewer 1atA**
>
> Thanks for your review that allows us to improve our paper.
>
> >Q1: what exactly is the use of the per-client score sorting?
>
> - A1: For Weakness 1, which concerns the necessity of maintaining per-client score rankings and Question 1, we employ per-client score sorting to compute the Q-digest, enabling the calculation of local ranks to introduce client heterogeneity in controlling fairness constraints (Proposition 3.2).
>
> >W2: Since the distributed quantile learning algorithm is not a contribution of this paper, and the notation is hard to follow, it is hard to evaluate the contribution and the insight of this work.
>
> - A2: We apologize for the complexity in notation. This complexity arises because our paper simultaneously considers different clients, ranks, quantiles, and groups. We will make efforts to simplify it. For the contribution and insight, we added a comparison with FaiREE to Appendix E and highlight the differences:
>
> "Regarding the differences between FedFaiREE and FaiREE, several pivotal distinctions become evident. Primarily, FedFaiREE demonstrates superior adaptability for practical applications. Notably, it incorporates mechanisms to handle label shift scenarios, ensuring model robustness within such distributions, as elucidated in Section 5.1. Furthermore, it's worth noting that FedFaiREE extends considerations to encompass multiple sensitive groups and multiple labels, aligning more closely with practical real-world application scenarios, as discussed in Appendix D.
>
> Another critical difference lies in the setting: FaiREE operates in a centralized environment, assuming homogeneous data across all clients. In contrast, FedFaiREE is expressly tailored for decentralized settings, acknowledging client heterogeneity and effectively addressing the challenges stemming from diverse data distributions and sizes across clients. This tailored approach significantly enhances its adaptability and robustness across various scenarios.
>
> Lastly, while FaiREE relies on specific centralized quantile estimation methods, FedFaiREE adopts approximate quantiles. This adaptation not only facilitates adaptation to distributed data but also fortifies the method's robustness and adaptability.”

---

### Official Review · Reviewer_NaPJ · 2023-11-08

**Soundness:** 2 fair
**Presentation:** 2 fair
**Contribution:** 2 fair
**Rating:** 5
**Confidence:** 2

**Summary:**

The authors propose a fairness Post-hoc approach that estimates a decision threshold for a classifier scoring function per sensitive attribute in a federated setting. This is done by computing a quantile estimate of the score function in a distributed manner such that the decision of the classifier achieves the best error subject to satisfying a confidence upper bound on the desired level of fairness. Note that the classifier needs to have access to the sensitive attribute at inference time since the learned decision threshold depends on it.

**Strengths:**

The idea seems promising, well grounded theoretically and the experiments show good performance for the proposed solution.

**Weaknesses:**

I find the presentation a bit hard to follow and should be simplified. I think the paper is hard to follow in terms of notation and procedure. The clients step is clear and the overall goal of having one threshold per sensitive attribute based on the desired fairness level seems reasonable. Even though this means that the sensitive attribute needs to be accessible at inference time.
However, update on the server and related notation is hard to follow. For instance it seems that K (Eq.4) is obtained based on the desired probability of not satisfying the fairness tolerance (|DEOO|> \alpha in Prop 3.2). Then the final pair k_0,k_1 are chosen to minimize the misclassification error (Eq 5) from the set of K. However, it is not clear to me how do you derive a single threshold for the scoring function from k_0,k_1 which seem to be two vectors of size S.

Even though I find the approach interesting I think the paper should be improved in terms of clarity of presentation. At least I find that a better presentation would clarify the contribution and attract more attention.

**Questions:**

Is the minimization of Eq. 5 done in a greedy manner? Do you evaluate all of the possibilities in K?

What is the global rank of a rank (i.e., last line in Step 2 Section 3.1 )

How does your method connect with Hardt et al 2016 which is a centralized post-processing fairness technique that also relies on finding a threshold per sensitive group to satisfy a fairness criteria.

---

> ### Author Response · Authors · 2023-11-22
> **Response to Reviewer NaPJ**
>
> Thanks for your review and helping us improve our paper.
>
> >Q1:Is the minimization of Eq. 5 done in a greedy manner? Do you evaluate all of the possibilities in K?
> - A1: One of the simplest methods involves constructing K is by exhaustively searching for $k_0$ and $k_1$, which has computational complexity of $O(n^2)$. To further reduce the computational cost, in the Appendix C.1, we introduce a simplified approach. We efficiently search for $k_0$ and $k_1$ by utilizing the relationship between the optimal classifier's thresholds, $t_0$ and $t_1$. This significantly simplifies the construction of K in practice (complexity O($n^{*,1}$)).
>
> >Q2: What is the global rank of a rank (i.e., last line in Step 2 Section 3.1 )
> - A2: In Proposition 3.1 of Section 3.2 in the original paper, we defined the local rank **k**. For any global rank k, we consider the vector composed of its corresponding values at the respective ranks (rounded down) in each local dataset, denoted as **k**. In practice, due to the subsequent introduction of a distributed algorithm, **k** represents estimates of these local ranks.
>
> >Q3: How does your method connect with Hardt et al 2016 which is a centralized post-processing fairness technique that also relies on finding a threshold per sensitive group to satisfy a fairness criteria.
> - A3: Comparing to the method proposed in Hardt et al. 2016, we are similar in the sense that both methods utilize scores to search for group-wise different thresholds. However, the search method is different. First of all, our method differs in that it provides provable control of fairness metrics (like DEOO with alpha-tolorance). This allows for a better control in trade-off between fairness and accuracy. Secondly, besides the difference in the design for decentralized processing and handling client heterogeneity, the method proposed in Hardt et al. 2016 does not have rigorous fair guarantees under the small-sample and distribution-free setting.

---

### Official Review · Reviewer_XnRQ · 2023-11-09

**Soundness:** 1 poor
**Presentation:** 2 fair
**Contribution:** 1 poor
**Rating:** 6
**Confidence:** 4

**Summary:**

This work presents FedFaiREE, an extension of the FaiREE post-processing method for fair classification, and its formal guarantees to the federated learning settings, leveraging the Q-digest method. Akin to FaiREE, the proposed framework works with any score-based function and outputs the best threshold to correct fairness violations. The provided experimental results illustrate that the proposed method shows promising performance on two Adult and Compas datasets.

**Strengths:**

* Addressing group fairness in federated learning is a very important and currently popular problem. Most existing approaches are in-process methods and the proposed one is a post-process method that can be combined with other methods.

* The authors provided experiments with baselines with and without applying FedFaiREE and showed improvements in the final models' performance.

* The paper presents its ideas in a clear and easy-to-follow manner.

**Weaknesses:**

* I find the novelty and contributions of this work to be limited, given that the main objective, formal guarantees, and algorithm are very similar to [1]. Moreover, the algorithm operates under the assumption of full client participation, a condition that may not always align with reality in FL settings.

*  The proposed problem and guarantees rely on the assumption of binary target and attribute/group variables, which is a restrictive assumption. This raises questions about the actual utility of this approach and what are its guarantees in more realistic scenarios (e.g., multiple sensitive groups and multiclass problems). I note that extending the work to multiclass problems is also identified by the authors, but there are existing works that address multiple attributes for these fairness metrics, e.g., [3].

* The paper misses discussion and comparison to other works proposing the same idea -- i.e., how to optimize a fairness metric and produce results akin to centralized ML using the local information from clients (e.g., [2] and [4]). Also, while the authors briefly mention [5], [4] a more explicit discussion of these proposed method's conceptual differences would benefit the paper.

* The experimental section requires enhancements: (1) the paper performs experiments using only two datasets, (2) important experimental details are missing, (e.g., the number of clients used in the experiments, standard deviation for each result, Dirichlet distribution parameter values that were explored, what is the sensitive group for each dataset etc.), (3) the comparison to AFL which optimizes for client-fairness (i.e. a different fairness concept in FL) should be justified.


[1] Puheng Li, James Zou, and Linjun Zhang. Fairee: fair classification with finite-sample and distribution-free guarantee. In The Eleventh International Conference on Learning Representations, 2022.

[2] Papadaki, A., Martinez, N., Bertran, M., Sapiro, G., and Rodrigues, M. (2022). Minimax demographic group fairness in federated learning.

[3] Y. Zeng, H. Chen, and K. Lee (2021). Improving fairness via federated learning.


[4] Hu, S., Wu, Z. S., and Smith, V. (2022). Fair federated learning via bounded group loss.

[5] Yahya Ezzeldin, Shen Yan, Chaoyang He, Emilio Ferrara, and A. Avestimehr. Fairfed: Enabling
group fairness in federated learning. Proceedings of the AAAI Conference on Artificial Intelligence, 37:7494–7502, 06 2023

**Questions:**

**Major**
*  Can you please give more insights (than the ones at the beginning of section 3.2) on why and how the proposed method differs from [1]?

* How does this work compare with the related works [2],[3],[4],[5] mentioned above? Is the group fairness definition studied here different from [2] and [4]? Please also revise the related work.

* How does FedFaiREE perform for different fairness constraint parameter $\alpha$, different levels of data heterogeneity across clients and confidence level $\beta$? My understanding is that only $\alpha=0.1$ for the adult dataset, $\alpha=0.15$ for compas, and $\beta=0.95$ for all experiments.

*  I'm interested in understanding how heterogeneity and imbalancedness are introduced across clients for these datasets, using the Dirichlet distribution. Why the parameter for adult was set to 1 and for compas was set to 10? Additionally, what is the standard deviation for each result reported in the tables (both supplementary and main)? This should be included in the results.

* What's the number of clients studied per dataset? If the number is low you should empirically examine how this approach scales for a larger number of clients. To illustrate that, you can consider for example the  ACSIncome dataset, where the data are naturally noniid and partitioned into 50 states and Puerto Rico and treat each place as a client (i.e., 51 clients).


 **Minor:**

* It would be good for the authors to acknowledge the concerns regarding the COMPAS dataset within the fairness community.

* The margins around Figure 1 around the figure require editing. The main text touches on the figure's description.

---

> ### Author Response · Authors · 2023-11-22
> **(1/4) Response to Reviewer XnRQ**
>
> Thank you for your review and valuable feedback.
>
> >W1: I find the novelty and contributions of this work to be limited, given that the main objective, formal guarantees, and algorithm are very similar to [1]. Moreover, the algorithm operates under the assumption of full client participation, a condition that may not always align with reality in FL settings.
> >
> >Q1: Can you please give more insights (than the ones at the beginning of section 3.2) on why and how the proposed method differs from [1]?
>
> - A1: For Weakness 1 and Question 1, in light of your comment, we added the following discussion to Appendix E and highlight the differences:
>
> >Regarding the differences between FedFaiREE and FaiREE[1], several pivotal distinctions become evident. Primarily, FedFaiREE demonstrates superior adaptability for practical applications. Notably, it incorporates mechanisms to handle label shift scenarios, ensuring model robustness within such distributions, as elucidated in Section 5.1. Furthermore, it's worth noting that FedFaiREE extends considerations to encompass multiple sensitive groups and multiple labels, aligning more closely with practical real-world application scenarios (see more details in Appendix D).
> >
> >Another critical difference lies in the setting: FaiREE operates in a centralized environment, assuming homogeneous data across all clients. In contrast, FedFaiREE is expressly tailored for decentralized settings, acknowledging client heterogeneity and effectively addressing the challenges stemming from diverse data distributions and sizes across clients. This tailored approach significantly enhances its adaptability and robustness across various scenarios.
> >
> >Lastly, while FaiREE relies on specific centralized quantile estimation methods, FedFaiREE adopts approximate quantiles. This adaptation not only facilitates adaptation to distributed data but also fortifies the method's robustness and adaptability.’’
>
> At the same time, regarding the label shift scenario in Section 5.1, it can be understood as a guarantee for a client that has not participated in our training.
>
> >W2: The proposed problem and guarantees rely on the assumption of binary target and attribute/group variables, which is a restrictive assumption. This raises questions about the actual utility of this approach and what are its guarantees in more realistic scenarios (e.g., multiple sensitive groups and multiclass problems). I note that extending the work to multiclass problems is also identified by the authors, but there are existing works that address multiple attributes for these fairness metrics, e.g., [3].
>
> - A2: We appreciate your suggestion, and we have included a discussion on multiple sensitive groups and multiple labels in Appendix D.
>
> In the case of multiple sensitive groups, referring to the DEOO concept (defined in equation (1) in the paper), we primarily consider the maximum value of DEOO between these protected groups and the unprotected residual portions to control the overall relaxation of Equality of Opportunity (see Definition D.1 for the multi-group version). At this point, similar to previous notions, we introduce fairness control by simultaneously managing multiple DEOOs and ensuring accuracy by incorporating considerations of $p_a$ and $p_{Y,a}$ across multiple groups.
>
> For the multi-label situation, we are inspired by [6] and consider the scenario where labels are m-dimensional vectors, where each component is binary. The results in Sections 3 and 4 of the article can be viewed as the case when m=1. In extending the one-dimensional results to higher dimensions, the main differences are as follows:
>
> 1. The different dimensions of labels may not be independent, hence, we introduce conditional distribution to address this. We estimate the second dimension based on the first dimension, and the third dimension based on both the first and second dimensions. This process is repeated for the remaining dimensions.
>
> 2. Q-digest cannot handle estimation of multi-dimensional distributions; therefore, we consider an iterative version of q-digest. First, we construct a q-digest based on the first dimension, then on the leaf nodes, we build q-digest based on the second dimension (the estimation error of quantiles in different dimensions differs).
>
>
> 3. In estimating accuracy, when extending from one dimension to higher dimensions, a closed interval in one dimension may no longer be a closed interval. Therefore, we have to consider its projection onto a closed interval in one dimension among the m dimensions.

---

> ### Author Response · Authors · 2023-11-22
> **(2/4) Response to Reviewer XnRQ**
>
> >W3: The paper misses discussion and comparison to other works proposing the same idea -- i.e., how to optimize a fairness metric and produce results akin to centralized ML using the local information from clients (e.g., [2] and [4]). Also, while the authors briefly mention [5], [4] a more explicit discussion of these proposed method's conceptual differences would benefit the paper.
> >
> >Q2: How does this work compare with the related works [2],[3],[4],[5] mentioned above? Is the group fairness definition studied here different from [2] and [4]? Please also revise the related work.
>
> - A3: Concerning Weakness 3 and Question 2, we are grateful for your insights, and we have made modifications in the related work section, added discussion of comparison in Appendix E.1 and added discussion of the fairness metric in Appendix B.3.
>
> More specifically, in Appendix B.3, we add:
>
> >**Definition** B.7
> A classifier h satisfies Bounded Group Loss (BGL) at level $\zeta$ under distribution $\mathcal{D}$ if for all $a \in A$, we have $\mathbb{E}[l(h(x), y) \mid A=a] \leq \zeta$.
> >
> >**Definition** B.8
>     A classifier $h$ satisfies Conditional Bounded Group Loss (CBGL) for $y \in Y$ at level $\zeta_y$ under distribution $\mathcal{D}$ if for all $a \in A$, we have $\mathbb{E}[l(h(x), y) \mid A=a, Y=y] \leq \zeta_y$.
> >
> >When considering y as a binary variable and the loss function l being the 0-1 loss function, BGL is equivalent to
> >$$\mathbb{P}[\hat{y}\neq y |\mid A=a] \leq \zeta,$$
> >holding for any a, whereas Demographic Parity refers to
> >$$\mathbb{P}[\hat{y}\neq y |\mid A=0]=\mathbb{P}[\hat{y}\neq y |\mid A=1].$$
> >In this context, BGL can be understood as a relaxation of Demographic Parity.
> >
> >Similarly, when considering y as a binary variable and the loss function l being the 0-1 loss function, CBGL is equivalent to
> >$$\mathbb{P}[\hat{y}\neq y |\mid A=a, Y=y] \leq \zeta_y,$$
> >holding for any a, whereas Equalized Odds refers to
> >$$\mathbb{P}[\hat{y}\neq y |\mid A=0,Y=y]=\mathbb{P}[\hat{y}\neq y |\mid A=1,Y=y].$$
> >In this context, CBGL can be understood as a relaxation of Equalized Odds.
>
> Similarly, by **Definition** B.9，the metric that [2] considers can also be understood as a relaxation of Demographic Parity in the context of considering y as a binary variable and the loss function l being the 0-1 loss function. Due to the complexity of the formula, we do not present it here.
>
> In Appendix E.1, we add:
> >Differences between FedFaiREE and other fair federated learning methods lie in their approach to addressing fairness concerns. Many methods, akin to this paper, extend the principles of centralized machine learning to decentralized settings, such as FedFB[3], FedMinMax[2], PFFL[4], and others. These methods primarily focus on introducing fairness penalties in the objective functions and incorporate client reweighting schemes and terms (in objective functions) reweighting schemes that consider global or local fairness. The key divergence between our approach and these methods is that the latter typically converge and provide fairness guarantees only in large-sample scenarios, lacking assurances for fairness in small-sample situations, especially under distribution-free assumptions. Empirical results from Table 1 in this paper demonstrate that compared to FedFaiREE, methods like FedFB, FairFed[5] are not as effective in controlling fairness in small-sample scenarios. Furthermore, as these methods are predominantly in-processing techniques, while FedFaiREE falls under post-processing methods, there is a potential for further integration to achieve improved fairness guarantees as shown in our experiments. Moreover, another significant characteristic of FedFaiREE is its capability to adjust the trade-off between fairness and accuracy according to specific fairness constraints. This control capacity has been demonstrated in numerous experiments, showcasing an ability that other methods lack.

---

> ### Author Response · Authors · 2023-11-22
> **(3/4) Response to Reviewer XnRQ**
>
> >W4: The experimental section requires enhancements: (1) the paper performs experiments using only two datasets, (2) important experimental details are missing, (e.g., the number of clients used in the experiments, standard deviation for each result, Dirichlet distribution parameter values that were explored, what is the sensitive group for each dataset etc.), (3) the comparison to AFL which optimizes for client-fairness (i.e. a different fairness concept in FL) should be justified.
> >
> >Q4: I'm interested in understanding how heterogeneity and imbalancedness are introduced across clients for these datasets, using the Dirichlet distribution. Why the parameter for adult was set to 1 and for compas was set to 10? Additionally, what is the standard deviation for each result reported in the tables (both supplementary and main)? This should be included in the results.
> >
> >Q5: What's the number of clients studied per dataset? If the number is low you should empirically examine how this approach scales for a larger number of clients. To illustrate that, you can consider for example the ACSIncome dataset, where the data are naturally noniid and partitioned into 50 states and Puerto Rico and treat each place as a client (i.e., 51 clients).
>
> - A4: Addressing Weakness 4, Question 4, and Question 5:
>     1. Due to time constraints, we have only incorporated another dataset, ACSIncome. We will include more datasets, including CelebA, Bank, and Dutch as suggested in our final version.
>     2. Thank you for your suggestion. We have updated Table 2 to include the number of clients,Table 4, 5 to reflect the standard deviations of metrics and added Table 6 to reflect standard deviations of metrics presented in Tables 1 (due to the space limitation). The sensitive group for each dataset is outlined in the dataset information in the EXPERIMENT Section. To enhance clarity, we have also included it in Table 2. The choice of the Dirichlet distribution parameter is based on available set segmentation requirements (ensuring a nonempty test set). We also conducted an experiment using a different Dirichlet distribution parameter on Adult as it shows in Appendix C.3.
>     3. Although AFL is a client-fairness method, our intention in using it is for an initialization of our proposed FedFaiREE method. It serves as an input to demonstrate the broad applicability of FedFaiREE that it can take any method as input, and ensure the resulting classifier, after post-processing, satisfies fairness under the federated learning setting even with finite-sample.
>
>
> >Q3: How does FedFaiREE perform for different fairness constraint parameter
> , different levels of data heterogeneity across clients and confidence level
> ? My understanding is that only $\alpha=0.1$ for the adult dataset, $\alpha=0.15$ for compas, and $\beta=0.95$ for all experiments.
>
> - A5: Regarding Question 3, we have conducted experiments with different fairness constraint parameters $\alpha$ and various confidence levels $\beta$ in Appendix C.3.
>
> Table R1: Results with different $\alpha$ on Compas using FedAvg as input. $\overline{ACC}$ and $\overline{|DEOO|}$ represent the averages of accuracy and DEOO (defined in Equation (1) in the paper). Std ACC and Std DEOO represent standard deviations of accuracy and DEOO.  $|DEOO|_{95}$ represents the 95\% quantile of DEOO since we set the confidence level of FedFaiREE to 95\% in our experiments.
> |       $\alpha$        | 0.13   | 0.14   | 0.15   | 0.16   | 0.17   | 0.18   | FedAvg  |
> |---------------|--------|--------|--------|--------|--------|--------|---------|
> | $\overline{ACC}$ | 0.658  | 0.658  | 0.659  | 0.659  | 0.659  | 0.659  | 0.662   |
> | Std ACC  | 0.011  | 0.010  | 0.010  | 0.011  | 0.011  | 0.011  | 0.011   |
> | $\overline{\|DEOO\|}$     | 0.046  | 0.049  | 0.051  | 0.055  | 0.062  | 0.62  | 0.126   |
> | Std DEOO      | 0.037  | 0.042  | 0.044  | 0.045  | 0.047  | 0.048  | 0.056   |
> | $\|DEOO\|_{95}$       | 0.118  | 0.125  | 0.137  | 0.156  | 0.158  | 0.158  | 0.223   |
>
> Table R2: Results with different $\beta$ on Compas using FedAvg as input. The notation meanings align with those in Table R1.
> |       $\beta$       | 0.01   | 0.03   | 0.05   | 0.07   | 0.09   | 0.11   | 0.13   | FedAvg |
> |--------------|--------|--------|--------|--------|--------|--------|--------|--------|
> | $\overline{ACC}$| 0.658  | 0.658  | 0.659  | 0.659  | 0.659  | 0.659  | 0.659  | 0.662  |
> | Std ACC | 0.010  | 0.010  | 0.010  | 0.010  | 0.010  | 0.010  | 0.010  | 0.011  |
> | $\overline{\|DEOO\|}$   | 0.046  | 0.050  | 0.051  | 0.052  | 0.053  | 0.055  | 0.055  | 0.126  |
> | Std DEOO     | 0.038  | 0.042  | 0.044  | 0.045  | 0.045  | 0.044  | 0.045  | 0.056  |
> | $\|DEOO\|_{95}$      | 0.118  | 0.125  | 0.137  | 0.148  | 0.156  | 0.156  | 0.156  | 0.223  |

---

> ### Author Response · Authors · 2023-11-22
> **(4/4) Response to Reviewer XnRQ**
>
> >Minor Q1: It would be good for the authors to acknowledge the concerns regarding the COMPAS dataset within the fairness community.
>
> - A6: For “minor Question 1”, we have addressed this by including the following information in Appendix C.2:
>
> >“We also note that there are concerns raised by the fairness community regarding the COMPAS dataset underscore crucial complexities within algorithmic fairness research. While Risk Assessment Instrument (RAI) datasets like COMPAS serve as prevalent benchmarks, their oversimplification of the intricate dynamics within real-world criminal justice processes poses significant challenges. Measurement biases and errors inherent in pretrial RAI datasets limit the direct translation of fairness claims to actual outcomes within the criminal justice system. Additionally, the technical focus on these data as a benchmark sometimes ignores the contextual grounding necessary for working with RAI datasets. Ethical reflection within socio-technical systems further highlights the necessity of acknowledging and grappling with the limitations and complexities inherent in RAI datasets.”
>
> >Minor Q2: The margins around Figure 1 around the figure require editing. The main text touches on the figure's description.
>
> - A7: Thank you for your comment, we have revised this in our paper.
>
> Once again, thank you for your thorough review and guidance, which have significantly contributed to the improvement of our paper.
>
> [1] Puheng Li, James Zou, and Linjun Zhang. Fairee: fair classification with finite-sample and distribution-free guarantee. In The Eleventh International Conference on Learning Representations, 2022.
>
> [2] Papadaki, A., Martinez, N., Bertran, M., Sapiro, G., and Rodrigues, M. (2022). Minimax demographic group fairness in federated learning.
>
> [3] Y. Zeng, H. Chen, and K. Lee (2021). Improving fairness via federated learning.
>
> [4] Hu, S., Wu, Z. S., and Smith, V. (2022). Fair federated learning via bounded group loss.
>
> [5] Yahya Ezzeldin, Shen Yan, Chaoyang He, Emilio Ferrara, and A. Avestimehr. Fairfed: Enabling group fairness in federated learning. Proceedings of the AAAI Conference on Artificial Intelligence, 37:7494–7502, 06 2023
>
> [6] Tianci Liu, Haoyu Wang, Yaqing Wang, Xiaoqian Wang, Lu Su, and Jing Gao. Simfair: A unified framework for fairness-aware multi-label classification. Proceedings of the AAAI Conference on Artificial Intelligence, 37(12), 14338-14346, 06 2023.

---

> > ### Comment · Reviewer_XnRQ · 2023-11-23
> >
> > I thank the authors for their response. I have revised my score accordingly.

---

### Author Response · Authors · 2023-11-22
**Summary of Paper Revision**

We appreciate the feedback from all reviewers, which has significantly contributed to the enhancement of our paper. The newly uploaded version includes the following key changes:

1. We added results for generalizing our method to the setting of multiple sensitive groups and multiple labels in Appendix D.
2. We added more discussions for differences between our proposed method and FaiREE in Appendix E.
3. We added more experiments involving different fairness constraint parameters $\alpha$ , various confidence levels $\beta$ and different parameters of Dirichlet distribution in Appendix C.3. This showcases corresponding changes in fairness metrics and accuracy.
4. We added more experimental results on a new dataset, ACSIncome in Appendix C.5.
5. We added more experimental results, including client quantities, standard deviation of accuracy, standard deviation of DEOO, standard deviation of DPE, in Appendix C.2, C.3, and C.4 respectively.
6. We added more discussions with the suggested related work in Section 1.1, Appendix B.3 and Appendix E.1.
7. We corrected all the typos suggested by reviewers.

We hope these changes could address the concerns of reviewers and improve the overall quality of the paper.

---

### Meta-Review · Area_Chair_nCap · 2023-12-11

**Metareview:**

This paper proposes a post-processing fairness mitigation algorithm for the federated setting which targets applications with small numbers of samples and no distributional assumption. The framework extends the recent FaiREE algorithm developed for use in the centralized setting. The experimental results demonstrate that the method improves the two group fairness metrics considered on several data sets, but the readability of the paper is reduced by complicated notation. The technical contribution of the paper is limited, as the objectives, guarantees, and algorithm are very similar to those from FaiREE, with modifications such as the use of a distributed quantile estimation algorithm to account for the federated setting.

**Justification For Why Not Higher Score:**

The technical contribution is limited.

**Justification For Why Not Lower Score:**

N/A

---

### Decision · Program_Chairs · 2024-01-16

Reject